# A large-scale study across the avian clade identifies ecological drivers of neophobia

ManyBirds Project[¶], Rachael Miller[1,2,3☯‡*], Vedrana Šlipogor[4,5,6☯],
Kai R. Caspar[7,8,9☯], Jimena Lois-Milevicich[10☯], Carl Soulsbury[11☯], Stephan A. Reber[12☯],
Claudia Mettke-Hofmann[13☯], Megan Lambert[14☯‡*], Benjamin J. Ashton[15],
Alice M. I. Auersperg[14], Melissa Bateson[16], Solenne Belle[17], Boris Bilčík[18],
Laura M. Biondi[19], Francesco Bonadonna[17], Desiree Brucks[20,21], Michael W. Butler[22],
Samuel P. Caro[17], Marion Charrier[23], Tiffany Chatelin[23], Johnathan Ching[24],
Nicola S. Clayton[1], Benjamin J. Cluver[22], Ella B. Cochran[25], Francesca Cornero[1],
Emily Danby[1], Samara Danel[17,26], Martina Darwich[27], James R. Davies[1,28],
Alicia de la Colina[29], Dominik Fischer[30,31], Ondřej Fišer[6], Florencia Foitzick[10],
Edward C. Galluccio[32], Clara Garcia-Co[33], Elias Garcia-Pelegrin[34,35], Isabelle George[23,36],
Kai-Philipp Gladow[37], Raúl O. Gómez[38], Anna Grewer[39], Katie Grice[40], Lauren M. Guillette[41],
Devon C. Hallihan[22], Katie J. Harrington[14], Frauke Heer[30], Chloe Henry[23],
Vladimira Hodova[18], Marisa Hoeschele[27], Cécilia Houdelier[23], Paula Ibáñez de Aldecoa[42,43],
Oluwaseun Serah Iyasere[21,44], Yuka Kanemitsu[45], Mina Khodadadi[46], Duc Khong[24],
Melanie G. Kimball[25], Ariana N. Klappert[47], Lucy N. Koch[14,31], Uta U. König von Borstel[20],
Lubor Košťál[18], Anastasia Krasheninnikova[48,49], Lubica Kubikova[18], Connor T. Lambert[41],
Daan W. Laméris[33,50], Courtenay G. Lampert[22], Oceane Larousse[51], Christine R. Lattin[25],
Zhongqiu Li[52], Michael Lindenmeier[47], Delia A. Lister[53], Julia A. Mackenzie[2], Selina Mainz[54],
Danna Masri[25], Jorg J. M. Massen[55,56], Laurenz Mohr[54], Wendt Müller[33], Paul M. Nealen[57],
Andreas Nieder[58], Aurèle Novac[59], Nínive Paes Cavalcante[10,60], Kristina Pascual[16],
Carla Pascual-Guàrdia[59], Ayushi Patel[25], Katarína Pichová[18], Cristina Pilenga[61],
Laurent Prétôt[62], John L. Quinn[40,63], Elena Račevska[64], Juan C. Reboreda[10],
Sam Reynolds[2], Amanda R. Ridley[32], Theresa Rössler[14,42], Francisco Ruiz-Raya[65,66],
Marina Salas[50], Beatriz C. Saldanha[67,68], Sebastián M. Santiago[10,60], Nikola Schlöglová[14],
Gia Seatriz[24], Eva Serrano-Davies[69], Eva G. Shair Ali[14], Janja Sirovnik[70], Zuzana Skalná[18],
Katie E. Slocombe[71], Masayo Soma[45], Tiziana Srdoc[72], Stefan Stanescu[41],
Michaela Syrová[6], Alex H. Taylor[73,74,75], Christopher N. Templeton[24,76],
Karlie Thompson[57], Sandra Trigo[67,68], Camille A. Troisi[23,77,78], Utku Urhan[69,79],
Maurice Valbert[49], Kees van Oers[69], Alberto Velando[66], Frederick Verbruggen[77],
Jorrit W. Verkleij[55], Alizée Vernouillet[77], Jonas Verspeek[33,50], Petr Veselý[6],
Auguste M. P. von Bayern[48,49], Eline Waalders[2], Benjamin A. Whittaker[41],
Ella R. Williamson[71], Vanessa A. D. Wilson[59,80,81], Michelle A. Winfield[24], Neslihan Wittek[46],
Karen K. L. Yeung[41], Jade A. Zanutto[55,82]

1 Department of Psychology, University of Cambridge, Cambridge, United Kingdom, 2 School of Life Sciences, Anglia Ruskin University, Cambridge, United Kingdom, 3 Faculty of Environment, Science and Economy, University of Exeter, Exeter, United Kingdom, 4 Department of Ecology and Evolution, University of Lausanne, Lausanne, Switzerland, 5 The Sense Innovation and Research Centre, Lausanne & Sion, Lausanne, Switzerland, 6 Department of Zoology, University of South Bohemia, České Budějovice, Czech Republic, 7 Department of Cell Biology, Heinrich-Heine-University Düsseldorf, Düsseldorf, Germany, 8 Department of Game Management and Wildlife Biology, Czech University of Life Sciences, Prague, Czech Republic, 9 Zoo Frankfurt, Frankfurt, Germany, 10 Departamento de Ecología, Genética y Evolución & IEGEBA—CONICET, Facultad de Ciencias Exactas y Naturales, Universidad de Buenos Aires, Buenos Aires, Argentina, 11 School of Natural Sciences, University of Lincoln, Lincoln, United Kingdom, 12 Department of Cognitive Science, Lund University, Lund, Sweden, 13 School of Biological and Environmental Sciences, Liverpool John Moores University, Liverpool, United Kingdom, 14 Messerli Research Institute, Department of Interdisciplinary Life Sciences, University of Veterinary Medicine Vienna, University of Vienna, Medical University of Vienna, Austria, 15 College of Science and Engineering, Flinders University, Adelaide, Australia, 16 Biosciences Institute, Newcastle University, Newcastle upon Tyne, United Kingdom, 17 CEFE, Univ Montpellier, CNRS, EPHE, IRD, Montpellier, France, 18 Institute of Animal

**Data availability statement:** The full data set (including species with 600-second trial cut off), final analysis data set (removing species with 600-second trial cut off) and R coding script are available on: Figshare https://doi.org/10.6084/m9.figshare.27324972. We have also included the numerical values and R coding scripts underlying all relevant figures (available via this Figshare link).

**Funding:** Funding listed directly supported contributions to this study through individually acquired awards: FWF START Project Y 1309 to AA, Consejo Nacional de Investigaciones Científicas y Técnicas (PIP 2022-2024 GI 11220210100521CO) to *ROG, Natural Sciences and Engineering Council of Canada NSERC RGPIN-2019-04733, NSERC Discovery Grant Launch Supplement, University of Alberta Faculty of Science Start-up Grant to LMG, National Science Foundation IOS-2237423 to *CRL, Beca Aves Argentinas 2022 to *JLM, Research Support Development Grant, Anglia Ruskin University to *RM, Swedish Research Council (Vetenskapsrådet) grant 2021-01650 to *SAR, Agencia Nacional de Promoción de la Investigación, el Desarrollo Tecnológico y la Innovación PICT 2019-00381 to JCR, National Science Foundation IOS: 2207395 to *CNT, Marie Skłodowska-Curie Action fellowship 'UrbanCog' Project No. 101062662 under the European Union's Horizon Europe Programme (https://marie-sklodowska-curie-actions.ec.europa.eu) to *CAT, Swedish Research Council (Vetenskapsrådet) International postdoc grant (2020-00719), and DR. J.L. DOBBERKE FOUNDATION (Dobberke/2349/202115) to *UU, ERC Consolidator grant (European Union's Horizon 2020 research and innovation programme, grant agreement No. 769595), and Methusalem Project 01M00221 (Ghent University) to FV, BOF postdoc fellowship (#BOF.PDO.2021.0035.01) to *AV and Mitacs Globalink Research Internship 2022 Project ID: 25251 to KLY. The funders had no role in study design, data collection and analysis, decision to publish, or preparation of the manuscript. The authors who received a salary from one of the funders are donated with an * above and listed below. ROG and JCR are Research Fellows of the Consejo Nacional de Investigaciones Científicas y Técnicas (CONICET), from which they receive their salaries. CL received 1 month of summer salary from National Science

Biochemistry and Genetics, Centre of Biosciences SAS, Bratislava, Slovak Republic, **19** Instituto de Investigaciones Marinas y Costeras—CONICET, Universidad Nacional de Mar del Plata, Buenos Aires, Argentina, **20** Department of Animal Breeding and Genetics, University of Giessen, Giessen, Germany, **21** Department of Crop and Animal Sciences, Thaer-Institute of Agricultural and Horticultural Sciences, Humboldt-Universität zu Berlin, Berlin, Germany, **22** Department of Biology, Lafayette College, Easton, Pennsylvania, United States of America, **23** UMR6552-Ethologie Animale et Humaine, CNRS, Universite de Rennes, Universite Caen Normandie, Rennes, France, **24** Biology Department, Pacific University, Forest Grove, Oregon, United States of America, **25** Department of Biological Sciences, Louisiana State University, Baton Rouge, Louisiana, United States of America, **26** Department of Brain and Cognitive Sciences, University of Rochester, Rochester, New York, United States of America, **27** Acoustics Research Institute, Austrian Academy of Sciences, Vienna, Austria, **28** School of Biological Sciences, University of Bristol, Bristol, United Kingdom, **29** Departamento de Conservación e Investigación, Fundación Temaikèn, Buenos Aires, Argentina, **30** Zoo Wuppertal, Wuppertal, Germany, **31** Raptor Center & Wildlife Zoo Hellenthal, Hellenthal, Germany, **32** School of Biological Sciences, University of Western Australia, Perth, Australia, **33** Department of Biology, Antwerp University, Antwerp, Belgium, **34** Department of Psychology, National University of Singapore, Singapore, Singapore, **35** Mandai Wildlife Group, Singapore, Singapore, **36** Institut des Neurosciences Paris-Saclay, CNRS, Université Paris-Saclay, Saclay, France, **37** Department of Animal Behaviour, Bielefeld University, Bielefeld, Germany, **38** CONICET-Departamento de Biodiversidad y Biología Experimental, Facultad de Ciencias Exactas y Naturales, Universidad de Buenos Aires, Buenos Aires, Argentina, **39** Zoo Krefeld gGmbH, Krefeld, Germany, **40** School of Biological, Earth and Environmental Sciences, University College Cork, Cork, Ireland, **41** Department of Psychology, University of Alberta, Alberta, Canada, **42** Department of Behavioural and Cognitive Biology, University of Vienna, Vienna, Austria, **43** Institute for Advanced Study in Toulouse, University of Toulouse 1 Capitole, Toulouse, France, **44** Department of Animal Physiology, Federal University of Agriculture, Abeokuta, Nigeria, **45** Department of Biological Sciences, Hokkaido University, Sapporo, Japan, **46** Department of Biopsychology, Ruhr University Bochum, Bochum, Germany, **47** Zoological Research Museum Koenig, University of Bonn, Bonn, Germany, **48** Department of Behavioural Neurobiology, Max-Planck-Institute for Biological Intelligence, Seewiesen, Germany, **49** Max-Planck Comparative Cognition Research Station, Loro Parque Fundación, Tenerife, Spain, **50** Antwerp Zoo Centre for Research and Conservation, Royal Zoological Society of Antwerp, Antwerp, Belgium, **51** Faculté des Sciences, Université Paris-Saclay, Gif-sur-Yvette, France, **52** Lab of Animal Behavior & Conservation, School of Life Sciences, Nanjing University, Nanjing, China, **53** Department of Biology, Pittsburg State University, Pittsburg, Kansas, United States of America, **54** Department of General Zoology, University of Duisburg-Essen, Essen, Germany, **55** Animal Behaviour and Cognition Group, Department of Biology, Utrecht University, Utrecht, The Netherlands, **56** Avifauna Bird Park, Alphen a/d Rijn, The Netherlands, **57** Department of Biology, Indiana University of Pennsylvania, Indiana, Pennsylvania, United States of America, **58** Animal Physiology, Institute of Neurobiology, University of Tuebingen, Tuebingen, Germany, **59** Department of Comparative Cognition, University of Neuchâtel, Neuchâtel, Switzerland, **60** Instituto de Ciências Biológicas e da Saúde, Universidade Federal de Alagoas, Maceió, Alagoas, Brazil, **61** Zoomarine Italia, Rome, Italy, **62** Department of Psychology and Counseling, Pittsburg State University, Pittsburg, Kansas, United States of America, **63** Environmental Research Institute, University College Cork, Cork, Ireland, **64** Zagreb Zoo, Zagreb, Croatia, **65** School of Biodiversity, One Health & Veterinary Medicine, University of Glasgow, Glasgow, United Kingdom, **66** Centro de Investigación Mariña, Universidade de Vigo, Vigo, Spain, **67** CIBIO Centro de Investigação em Biodiversidade e Recursos Genéticos, InBIO Laboratório Associado, Universidade do Porto, Vairão, Portugal, **68** BIOPOLIS Program in Genomics, Biodiversity and Land Planning, Vairão, Portugal, **69** Department of Animal Ecology, Netherlands Institute of Ecology (NIOO-KNAW), Wageningen, The Netherlands, **70** Department of Farm Animals and Veterinary Public Health, University of Veterinary Medicine Vienna, Vienna, Austria, **71** Department of Psychology, University of York, York, United Kingdom, **72** Department of Philosophy, University of Vienna, Vienna, Austria, **73** Institut de Neurociències, Universitat Autònoma de Barcelona, Bellaterra, Barcelona, Spain, **74** ICREA, Barcelona, Spain, **75** School of Biological Sciences, University of Canterbury, Christchurch, New Zealand, **76** Department of Biology, Western Washington University, Bellingham, Washington, United States of America, **77** Centre for Research on Ecology, Cognition and Behaviour of Birds, Ghent University, Gent, Belgium, **78** Department of Experimental Psychology, Ghent University, Gent, Belgium, **79** Department of Biology, Lund University, Lund, Sweden, **80** Department of Comparative Language Science, University of Zurich, Zurich, Switzerland, **81** School of Psychology and Social Work, University of Hull, Hull, United Kingdom, **82** Faculté des Sciences et Ingénierie, Sorbonne Université, Paris, France

☙ These authors represent the ManyBirds Study 1 Leadership team (wider authorship team listed in alphabetical order).
‡ These two authors are the ManyBirds Project Founders and Project Leads.

Foundation IOS-2237423. JLM is supported by a postdoctoral fellowship from CONICET, through which she receives a stipend. RM received a salary as staff at Anglia Ruskin University. SAR received a salary from the Swedish Research Council. CNT received a salary from the National Science Foundation. CAT's salary was paid via ERC Consolidator grant (European Union's Horizon 2020 research and innovation programme, grant agreement No. 769595). AV received a salary through a BOF postdoc fellowship (#BOF.PDO.2021.0035.01) from the University of Ghent. UU received as salary from the Swedish Research Council (Vetenskapsrådet) International postdoc grant (2020-00719).

**Competing interests:** The authors have declared that no competing interests exist.

**Abbreviations**: DNH, Dangerous Niche Hypothesis; LHT, Life History Theory; NTH, Neophobia Threshold Hypothesis.

¶ Membership of the ManyBirds Project is listed in the Acknowledgments.
* rmam3@cam.ac.uk (RM); Megan.Lambert@vetmeduni.ac.at (ML)

## Abstract

Neophobia, or aversion to novelty, is important for adaptability and survival as it influences the ways in which animals navigate risk and interact with their environments. Across individuals, species and other taxonomic levels, neophobia is known to vary considerably, but our understanding of the wider ecological drivers of neophobia is hampered by a lack of comparative multispecies studies using standardized methods. Here, we utilized the ManyBirds Project, a Big Team Science large-scale collaborative open science framework, to pool efforts and resources of 129 collaborators at 77 institutions from 24 countries worldwide across six continents. We examined both difference scores (between novel object test and control conditions) and raw data of latency to touch familiar food in the presence (test) and absence (control) of a novel object among 1,439 subjects from 136 bird species across 25 taxonomic orders incorporating lab, field, and zoo sites. We first demonstrated that consistent differences in neophobia existed among individuals, among species, and among other taxonomic levels in our dataset, rejecting the null hypothesis that neophobia is highly plastic at all taxonomic levels with no evidence for evolutionary divergence. We then tested for effects of ecological factors on neophobia, including diet, sociality, habitat, and range, while accounting for phylogeny. We found that (i) species with more specialist diets were more neophobic than those with more generalist diets, providing support for the Neophobia Threshold Hypothesis; (ii) migratory species were also more neophobic than nonmigratory species, which supports the Dangerous Niche Hypothesis. Our study shows that the evolution of avian neophobia has been shaped by ecological drivers and demonstrates the potential of Big Team Science to advance our understanding of animal behavior.

## Introduction

Aversion to novelty, also known as neophobia, is often measured as the latency (i.e., time taken) to approach or interact with novel stimuli (e.g., objects or food) compared to familiar stimuli [1]. Neophobic responses can protect an individual from potential risks, but may also decrease opportunities to exploit novel resources, such as unknown food or nesting sites [2]. Thus, patterns of neophobic responses across taxa are likely to be driven by both benefits and costs [3], and these are linked to taxon-specific ecological and social factors [4]. Neophobia impacts adaptability and survival [5,6], and can affect ecological processes [7], including invasiveness (cane toads, *Rhinella marina* [8]; common mynas, *Acridotheres tristis* [9]), and range and niche expansion (small mammals [10]; Eurasian tree sparrows, *Passer montanus* [11]).

Individuals, populations, and species vary in their degree of neophobia [12,13]. For individuals, repeatability (i.e., consistency) in neophobic responses has been demonstrated in various clades (e.g., mammals [14], birds [15,16]). However, plasticity in

neophobia is also evident, such as when house sparrows (*Passer domesticus*) learn to be less neophobic from conspecifics [17] and rooks (*Corvus frugilegus*) overcome neophobia in groups [18]. Although neophobia can change on a within-individual level, most studies assessing neophobia find that it is fairly consistent within individuals when studied in the same context and developmental stage (but see conflicting data for corvids [18]) and varies widely among different species. Although the drivers of variation in neophobia are relatively unknown, there are at least two nonmutually exclusive hypotheses specifically addressing between-species variation in neophobia, which are inspired by Russell Greenberg's seminal studies [19–21]; see also [22]. The Neophobia Threshold Hypothesis [19,21,23] states that neophobia acts as a proximate mechanism for niche specialization, as it reduces the probability of a species exploring new stimuli in the environment, thereby shifting its diet and habitat use. This hypothesis is supported by findings that habitat generalists typically demonstrate lower neophobia than habitat specialists [19,23]. The Dangerous Niche Hypothesis [2,24] suggests that neophobia increases with the level of environmental danger and protects against risks, predicting species that utilize unfamiliar or riskier habitats (e.g., higher predator density) will show higher neophobia. In support of this hypothesis, fish and amphibians from environments with higher predation risk exhibit increased neophobia levels [25].

Neophobia may also be shaped by life history traits. Life History Theory predicts that species with slow life histories (e.g., long-life expectancy, long reproductive lifespans, slower growth) would be more risk-averse and thus neophobic, while species with fast life histories (e.g., short life expectancy, high fecundity, faster growth) would be likely risk-takers [26,27]. This framework may help to explain associations between neophobia and species traits such as body size and domestication status [28,29], although a more thorough test of the predictions of Life History Theory would require comparing neophobia in a wider range of taxa than has been done thus far.

Birds provide an excellent taxon in which to investigate the ecological drivers of neophobia. They vary widely in factors such as diet (including foraging behavior), habitat use diversity, body size, and sociality [30], thereby making them a suitable group for testing predictions of the Neophobia Threshold Hypothesis, Dangerous Niche Hypothesis, and Life History Theory (Table 1). Previous research on birds found that neophobic behaviors correlate with both dietary breadth [19,23,31,32] and degree of sociality [12]. The degree of ecological generalization has been found to display a negative correlation with neophobia in multiple bird taxa [20,32]. Similarly, sociality can also influence neophobia by moderating risk or the "dilution effect" (i.e., reduction in predation risk), as predicted by the Dangerous Niche Hypothesis; for instance, the presence of conspecifics can either enhance or reduce neophobia (e.g., zebra finches, *Taeniopygia guttata,* [12]; common ravens, *Corvus corax* and carrion crows, *Corvus corone,* [33]). In 10 species of corvids, for example, species living in family groups showed lower object neophobia than those living in territorial pairs [15].

Neophobia has also been linked to birds' ability to exploit novel habitat types such as urban environments, which would be in agreement with the Neophobia Threshold Hypothesis. Across more than 8,000 bird species, a phylogenetically controlled analysis showed that only 9% of species inhabit urban environments and are characterized by using a broad array of natural habitats [34]. Resources in urban environments may therefore represent an extension of the varied resources that a habitat generalist typically encounters in natural environments. At an individual level, common mynas living in urban environments showed lower neophobia towards novel food resources than those living in rural areas [35]. Across 12 (primarily passerine) bird species, individuals were quicker to approach human litter in an urban environment compared to a rural environment [36].

Despite evidence for intriguing links between variation in neophobia and different ecological factors, it is unknown how generalizable results are across avian taxa. Few studies have used consistent methods to compare neophobia across multiple bird species, and those that did only focused on closely related taxa (e.g., [19]). The most comprehensive study so far comparatively examined 61 parrot species and found that neophobia was positively related to an insect-rich diet and negatively related to a more leaf-based diet [13]. These results highlight that comparative analysis can provide a powerful tool for exploring the ecological drivers of behavior. Other large-scale multispecies studies have been pioneered through Big Team Science collaborative approaches, also known as 'ManyX' projects. Driven by early efforts from human

**Table 1. Ecological factors assessed for correlations with object neophobia in birds.**

| Category | Factor | Predictions | Coding and definition | Data source |
|---|---|---|---|---|
| Diet | **Dietary breadth** (i.e., feeding specialist vs. generalist) | Generalists are expected to show lower levels of neophobia, given lower constraints on foraging strategies (NTH). | Seven categories (coded: 1–7). Species coded on a scale from 1 (one food category consumed; specialist) to 7 (seven food categories consumed; generalist). | Wilman and colleagues [47] |
| Sociality | **Territoriality** | Less territorial species are expected to exhibit lower neophobia, given lower constraints on habitat use, i.e., less territorial species are "spatial generalists" compared to more territorial "spatial specialists" (NTH). | Three categories (coded: non-territorial, seasonally territorial, year-round territorial). | Tobias and colleagues [48] |
| Habitat and range | **Habitat complexity** (i.e., structural complexity) | Species occupying more complex habitats are expected to exhibit lower levels of neophobia, as effective foraging and habitat exploitation requires higher degrees of exploration (NTH). | Three categories (coded: 1–3). Structural habitat complexity ranged from 1 (high complexity, e.g., forest/dense shrubland) to 3 (low complexity, e.g., grasslands, deserts, or open water). | Tobias and colleagues [49] |
| | **Habitat use diversity** (i.e., more habitat specialist vs. generalist) | Habitat specialists are expected to show higher levels of neophobia than generalists because their habitat use patterns are more constrained (NTH). | Occupation of different habitat types (coded: 1–10 types; as defined by BirdLife International (Level 1 habitats), species coded on a scale from 1 (one habitat type used; specialist) to 10 (10 habitat types used; generalist). | BirdLife International [50] |
| | **Anthropogenically modified habitat use** (i.e., altered by human settlements and infrastructure, excluding plantation areas based on their Urban Association Index, UAI) | Species occupying anthropogenically modified habitats are expected to show lower levels of neophobia because they have to maneuvre dynamic conditions more frequently than nonurban inhabitants. Thus, low neophobia levels might represent an exaptation for occupying anthropogenically modified habitats (NTH). | Two categories (coded: urban avoider or exploiter). The global average UAI, derived from data on 3,768 bird species, is 1.14. Species coded as avoiders when UAI below 1.14 and exploiters if above. UAI was available for 89 species in our sample. For the remaining 47 species, we relied on additional primary literature and collective expertise of our collaborators to code them. | Neate-Clegg and colleagues [51]; Sol and colleagues [52] |
| | **Migratory habits** | Migratory and nomadic species are expected to be more neophobic compared to nonmigratory ones, as they move through various environments with limited opportunity to become familiar with local conditions. This might make novel stimuli and environmental change more likely to be perceived as risky (DNH). | Three categories—migratory (a significant part of the species' populations regularly travels beyond their breeding areas, following a predictable timing and specific routes each season), nomadic (populations are not resident but may constantly move following the availability of sporadic [in time and distribution] resources. In contrast to migratory species, routes and timing of movements cannot be adequately predicted in the long term), nonmigratory (movement patterns do not comply to definitions outlined above, resident populations). | IUCN [53] |
| Other | **Body mass** | Larger-bodied species should exhibit lower neophobia than smaller-bodied ones because they experience lower predation risk (DNH); alternatively, larger-bodied species may be more neophobic, as they also tend to have fewer offspring, longer lifespans, and longer generation times (LHT). | Continuous—species' average body mass in grams (g). | Tobias and colleagues [49] |
| | **Domestication status** | Domesticated lineages should exhibit lower neophobia than nondomesticated ones, given that unpredictable risks are largely absent from the artificial environments inhabited by domesticated birds (DNH) and domesticated animals are bred to have more offspring and shorter generation times (LHT). | Two categories—domesticated or nondomesticated. | Domestication status reported by respective collaborators |

All factors were coded at species level. For domesticated forms, information on the respective ancestral species from the wild was coded. Predictions in line with the Dangerous Niche Hypothesis (DNH), Neophobia Threshold Hypothesis (NTH), and Life History Theory (LHT) are annotated. Each factor was coded per species using or informed by peer-reviewed or reputable sources, where available (referenced in Data Source column). Full species coding sheet available in Table A in S1 Text.

psychology to address the replication crisis (Open Science Collaboration; [37,38]), other fields have followed suit, including comparative behavior and cognition (ManyPrimates: [39,40]; ManyBirds: [41]; ManyDogs: [42]; ManyManys: [43]). These collaborations allow researchers to ask questions that individual labs do not have the resources to address and provide large-scale replications of existing studies by facilitating more diverse species representation and larger sample sizes [44]. For example, in a review of avian cognition and behavior research from 2015 to 2020, Lambert and colleagues [41] identified that only ~1% of all bird species were represented in cognitive studies. The studies originated largely from four countries (UK, USA, Canada, and Austria) and represented only four taxonomic orders. Most (75%) subjects were from laboratory sites, with field sites and zoos being underrepresented. The present study sought to increase the representation of avian orders, species, and individuals to capture wider variation across the avian clade and examine generalisability, as well as include more researchers worldwide.

Using a Big Team Science approach, we aimed to test predictions of the Neophobia Threshold Hypothesis, Dangerous Niche Hypothesis, and Life History Theory [3] for avian neophobia, to establish whether neophobia in birds is a stable trait within individuals, and to explore its phylogenetic patterns. We assayed "object neophobia" (following terminology outlined in [45]) using a standardized approach for 1,439 subjects across 136 species and 25 taxonomic orders (Figs 1 and 2), as Study 1 of the ManyBirds Project (www.themanybirds.com; [41]). Data were collected from labs, zoos, and field sites to increase species representation, and because zoos in particular provide access to some rare species less represented in existing literature [46]. We measured latency to touch familiar food presented alongside a novel object and compared this with latency to touch the same familiar food without a novel object present. We analyzed both: (1) difference scores (novel object test minus control latency values) and (2) raw data (control and novel object test latencies separately). Difference scores provide the change between the two conditions and reflect the actual neophobia elicited, and are independent of the control measure, i.e., how much longer an individual waited to feed with the novel object present. However, where this difference occurs (e.g., at fast or slow latencies) and its context is functionally meaningful, which is captured by the raw data. We had two main objectives. The first was to examine whether neophobia was entirely plastic with no evidence of evolutionary divergence at any level or whether consistent differences in neophobia existed among individuals, among species, and among other taxonomic levels in our dataset. We then examined what ecological factors, including diet, sociality, habitat, and range-predicted neophobia, while accounting for phylogeny (Table 1).

For objective 1, we wanted to establish the extent to which there are consistent individual differences in neophobia by measuring repeatability, which sets the upper limit to heritability - an important component of the evolvability of a trait within a population. We hypothesized that neophobic responses are consistent across individuals and species; alternatively, responses may be highly plastic, with no consistent differences across individuals, species, or even higher taxonomic groups. We thus tested this hypothesis, noting that species may differ in baseline neophobia despite individual plasticity (i.e., neophobia is a species feature). The repeatability of behavior has been well studied and varies considerably depending on a diverse range of factors [56]. However, we expected that individuals would be largely repeatable in their neophobic responses, given the relatively short time intervals between different test rounds (~2 weeks), as suggested by a meta-analysis finding larger repeatabilities in short- than long-term object neophobia studies [45].

For objective 2, we explored phylogenetic patterns of variation and expected to find variation across avian orders in neophobia, given the vast diversity in social and ecological traits across the avian clade [41] and previously observed species differences in neophobia (e.g., parrots [13]; corvids [15]). Crucially, we then tested whether differences are driven by ecological factors, including diet, territoriality, habitat, and body size to influence neophobia, based on findings reported by single-species and smaller multispecies studies [14,57,58]. Specifically, we tested for differences in neophobia between species varying in migratory habits (i.e., migratory, nomadic, and nonmigratory species) and habitat exploitation, including feeding and habitat generalists versus specialists. Following the Dangerous Niche Hypothesis, we expected nonmigratory species to be less neophobic than migratory ones, as the latter evolved to maneuvre diverse environments but with limited

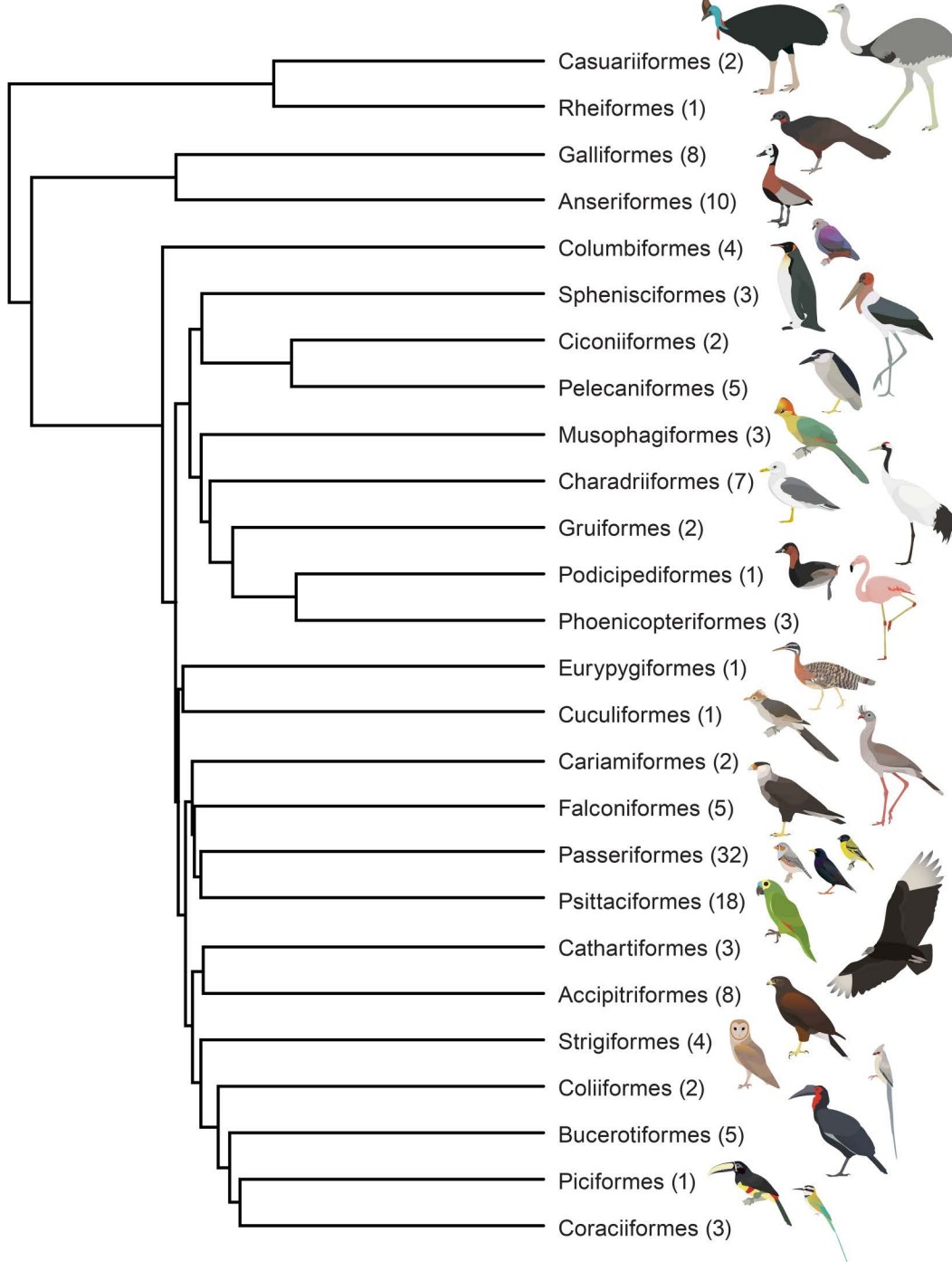

**Fig 1. Phylogeny of extant avian orders represented in ManyBirds study 1.** Numbers in parentheses indicate the number of species in the given order representing 136 species in total across 25 orders. Phylogeny based on Jetz and colleagues [54]. Bird drawings created by Raúl O. Gómez (coauthor) and shared with permission.

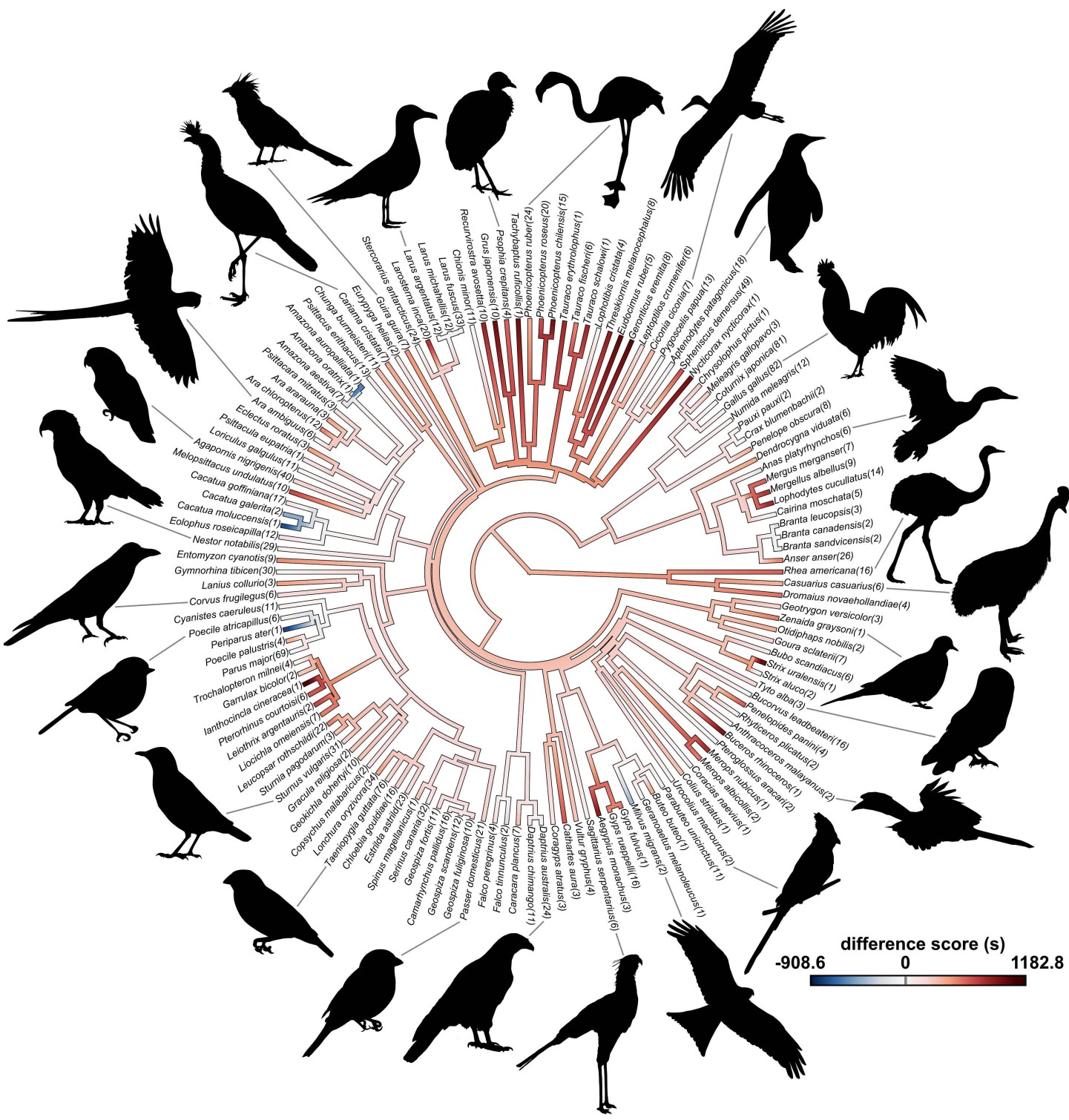

**Fig 2. Color-coded latency difference scores (latency to touch food in the novel object condition minus latency in the control condition; measured in seconds) for all bird species included in the analyses (*n* = 136).** Scores reconstructed for nodes represent maximum likelihood estimates. Sample sizes (total number of individuals and/or flocks tested) are provided in parentheses. Silhouettes are in the public domain and derive from PhyloPic (www.phylopic.org). The color-coded tree was created with the *contMap()* function from the *phytools* package [55]. The data underlying this figure can be found in https://doi.org/10.6084/m9.figshare.27324972.

opportunity to become familiar with local conditions [19,23] (Table 1). In line with the Neophobia Threshold Hypothesis [19,23], we predicted that neophobia would be lower in generalists than specialists, due to increased exposure to a wide variety of novel stimuli and environments (Table 1).

We also expected that social factors would influence neophobia, following previous findings in birds and other taxa (e.g., [12,14,33,59]). For example, more territorial species may be thought of as "spatial specialists", thereby showing higher neophobia in keeping with the Neophobia Threshold Hypothesis, while nonterritorial species with more fluid or overlapping home ranges may be "spatial generalists" showing lower neophobia. Other potential factors of notable influence include body size/mass and domestication. Body size has been linked to Life History Theory, and we thus expected larger birds to have a slower pace of life and to be more neophobic [60,61]. However, larger body size has also been linked to lower predation pressure, which is expected to result in lower neophobia in larger species, as per the Dangerous Niche Hypothesis [62]. In this case, the two hypotheses about the relationship between body size and neophobia are mutually exclusive, although it is also possible that different ecological drivers may shape neophobic responses in different avian families. With domestication, lower reactivity and reduced fear or stress responses are often considered desirable traits among domestic animals, allowing them to cope and reproduce more effectively in human-controlled environments [29, 63–65]. Domesticated animals are also bred to have more offspring and shorter generation times, and the anthropogenically modified environments they dwell in are safer than wild environments in terms of predation and food availability [66]. Therefore, both the Dangerous Niche Hypothesis and Life History Theory would predict that levels of neophobia should be lower in domesticated lineages compared to nondomesticated ones.

Ultimately, this study enabled us to identify ecological drivers of neophobia in the avian clade and to demonstrate the utility of the ManyBirds Project as a framework for future comparative studies on the evolution of phenotypic variation in birds.

## Results

### Individual repeatability

At the individual level within species, responses to novel stimuli (based on raw data) were contextually repeatable across control and novel object test conditions (repeatability estimate: $R = 0.443$ [0.41/0.48], $p < 0.001$). In addition, individual responses were temporally repeatable within each condition (control: $R = 0.429$ [0.38/0.48], $p < 0.001$; novel object test: $R = 0.521$ [0.48/0.56], $p < 0.001$). Repeatability was also found at the species, family and order level (e.g., species-level repeatability: control: $R = 0.332$ [0.252/0.398], $p < 0.001$; novel object test: $R = 0.499$ [0.417/0.565], $p < 0.001$; differences scores: $R = 0.391$ [0.310/0.459], $p < 0.001$; Table C in S1 Text). Furthermore, repeatability was found when all ecological and other factors were included as fixed effects in the models (control: $R = 0.48$ [0.43/0.52], $p < 0.001$; novel object test: $R = 0.61$ [0.57/0.65], $p < 0.001$; difference scores: $R = 0.27$ [0.21/0.32], $p < 0.001$).

### Ecological drivers of neophobia

Avian order affected control and novel test conditions, as well as difference score latencies (Table D in S1 Text, Fig A in S1 Text, and Information A in S1 Text). Across species, using individual difference scores (novel object test values minus control values, i.e., not raw data), dietary breadth and migratory pattern had an effect on neophobia for both Model A, which included the full data set (Table 2), and Model B, which included nondomesticated species only (Table E in S1 Text, removing seven domesticated lineages). In line with the Neophobia Threshold Hypothesis, species that consume fewer types of food (i.e., feeding specialists) had higher difference scores (i.e., higher object neophobia) than feeding generalist species with broader diets (Fig 3). In line with the Dangerous Niche Hypothesis, migratory species had higher difference scores (i.e., higher neophobia) than nonmigratory species (Fig 3). Social context had an effect for Model B only (Table E

**Table 2. Generalized linear mixed models using Markov chain Monte Carlo estimation methods (MCMCglmm models), testing the effect of predictors on birds' difference scores (novel object minus control values of latency to touch familiar food; Model A).**

| | Post.mean | L. CI | U. CI | Eff. samp | $p_{MCMC}$ |
|---|---|---|---|---|---|
| **(Intercept)** | 637.78 | −116.36 | 1399.96 | 1980 | 0.104 |
| Social Context | 21.90 | −27.65 | 75.23 | 1980 | 0.408 |
| Test Order (Novel Object Test Presented Second) | −31.41 | −66.48 | 6.86 | 1980 | 0.087 |
| Test Order (Both Same Day) | −15.82 | −141.01 | 115.53 | 1946 | 0.819 |
| Body Mass (log) | −5.26 | −66.99 | 51.48 | 1980 | 0.831 |
| **Dietary Breadth** | −62.48 | −108.95 | −13.43 | 1980 | **0.008** |
| Habitat use diversity | −3.80 | −42.06 | 39.13 | 1980 | 0.860 |
| Anthropogenic Habitat Use (Exploiter) | 96.60 | −12.18 | 211.49 | 1980 | 0.090 |
| habitat complexity | 57.53 | −42.14 | 162.14 | 1980 | 0.260 |
| Territoriality (Seasonal) | 17.93 | −183.91 | 209.24 | 1980 | 0.848 |
| Territoriality (Year-Round) | −81.31 | −309.77 | 140.71 | 1980 | 0.480 |
| Migration (Nomadic) | −163.87 | −539.12 | 226.23 | 1901 | 0.394 |
| **Migration (Nonmigrant)** | −136.15 | −275.15 | −6.37 | 1980 | **0.047** |
| Domestication Status | 11.89 | −207.27 | 231.09 | 1980 | 0.915 |

Individual ID, nested within site, and phylogeny were included in the model as random effects to control for nonphylogenetic and phylogenetic nonindependence, respectively, among individuals. $p_{MCMC}$ is twice the posterior probability that the estimate is negative or positive (whichever probability is smallest), L. CI = lower 95% credible interval, U. CI = upper 95% credible interval. For migratory patterns, the 0 intercept in this 3-level factor is migratory species. Dietary breadth, habitat complexity, and habitat use diversity were included in the model as continuous variables. Significant predictors ($p < 0.05$) are indicated in bold. Eff. samp is the effective sample size of a Monte Carlo computation.

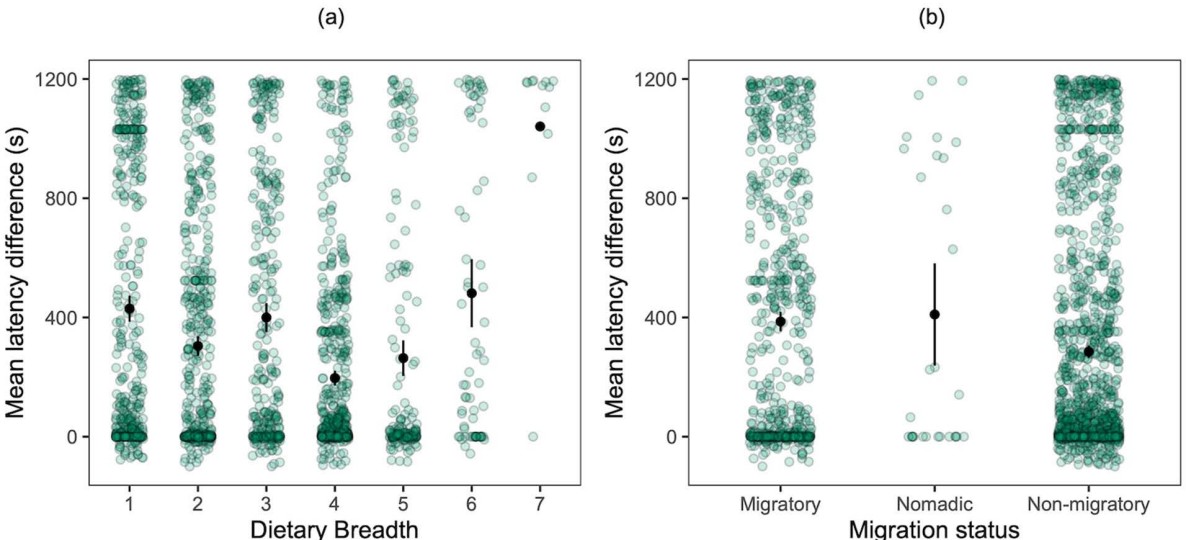

(a) (b)

**Fig 3. Mean ± SE latency difference scores (novel object minus control values) of birds to touch familiar food in relation to a) dietary breadth and b) migratory pattern.** A higher difference score indicates a (higher) neophobic response to novel object presence. Circles represent individual difference scores. For a) dietary breadth, note that categories 1 to 5 represent 133 species, thus driving this effect (categories 6 and 7 contain two and one species, respectively). The data underlying this figure can be found in https://doi.org/10.6084/m9.figshare.27324972.

in S1 Text), with higher neophobic responses (longer latencies) when birds were tested in a social context (108 species) than when alone (58 species). Test order had no effect on neophobia for Model A or B.

Using raw data (control and novel object test latencies separately; Model C), the addition of a novel object overall increased the latency of birds to touch familiar food (Table 3). Neophobia conditions interacted with: (1) dietary breadth, (2) migratory patterns, (3) habitat use diversity, (4) habitat complexity, (5) domestication status, (6) test order, and (7) social context. The dietary breadth and migratory patterns findings reported for the difference scores (Fig 3) were also recorded for the raw data (Figs 4 and 5). No effects of territoriality, anthropogenically modified habitat use, or body mass were detected. Therefore, under novel object conditions, species that were more restricted (versus broader) in dietary breadth (Fig 4), migratory (versus nonmigratory; Fig 5), had low habitat complexity (versus high habitat complexity; Fig B in S1 Text), more restricted (versus broader) in habitat use diversity (Fig C in S1 Text) and were nondomesticated (versus

**Table 3. Generalized linear mixed models using Markov chain Monte Carlo estimation methods (MCMCglmm models), testing the effect of predictors on the raw data of birds' latency to touch familiar food in novel object test and control trials (Model C).**

|  | Post.mean | L. CI | U. CI | Eff. samp | $p_{MCMC}$ |
|---|---|---|---|---|---|
| (Intercept) | 253.09 | −241.89 | 787.72 | 2166.000 | 0.355 |
| Condition (Novel Object Test/Control) | 25.71 | −97.67 | 161.04 | 1980.000 | 0.723 |
| **Social Context** | 162.52 | 120.23 | 207.70 | 1980.000 | **<0.001** |
| **Trial Order (Second Trial)** | 32.38 | 4.51 | 64.71 | 2712.000 | **0.038** |
| Trial Order (Both Same Day) | −11.85 | −102.96 | 79.05 | 2161.000 | 0.800 |
| Body Mass (log, grams) | −26.08 | −69.77 | 12.08 | 2469.000 | 0.205 |
| Dietary Breadth | 20.11 | −14.15 | 52.86 | 1980.000 | 0.247 |
| habitat complexity | −12.72 | −85.23 | 58.98 | 2002.000 | 0.731 |
| Anthropogenic Habitat Use (Exploiter) | 13.46 | −71.11 | 103.37 | 1980.000 | 0.737 |
| Habitat use diversity | 26.39 | −2.68 | 53.17 | 1980.000 | 0.071 |
| Territoriality (Seasonal) | −39.72 | −172.62 | 110.56 | 1980.000 | 0.573 |
| Territoriality (Year-Round) | −20.13 | −178.27 | 140.09 | 1980.000 | 0.816 |
| Migration (Nomadic) | 115.54 | −158.40 | 388.30 | 1980.000 | 0.428 |
| **Migration (Nonmigrant)** | −103.89 | −204.33 | −10.96 | 1980.000 | **0.039** |
| Domesticated | 5.50 | −146.41 | 162.12 | 1980.000 | 0.951 |
| **Condition × Social Context** | 155.71 | 118.02 | 198.19 | 2297.000 | **<0.001** |
| Condition × Trial Order (Second Trial) | 4.27 | −44.65 | 52.39 | 1980.000 | 0.893 |
| Condition × Trial Order (Both Same Day) | 61.73 | −17.42 | 152.96 | 2144.000 | 0.152 |
| Condition × Body Mass | 8.64 | −2.11 | 19.08 | 1980.000 | 0.113 |
| **Condition × Dietary Breadth** | −29.57 | −43.95 | −14.60 | 1980.000 | **0.001** |
| **Condition × Habitat complexity** | 41.19 | 9.30 | 75.71 | 1857.000 | **0.012** |
| Condition × Anthropogenic Habitat Use (Exploiter) | −27.36 | −72.57 | 19.97 | 2154.000 | 0.253 |
| **Condition × Habitat use diversity** | 12.43 | 0.82 | 25.66 | 1762.000 | **0.049** |
| Condition × Territoriality (Seasonal) | 41.37 | −26.11 | 104.56 | 2358.000 | 0.214 |
| Condition × Territoriality (Year-Round) | −40.92 | −104.01 | 22.03 | 1980.000 | 0.197 |
| Condition × Migration (Nomadic) | −68.02 | −221.16 | 87.40 | 1980.000 | 0.385 |
| Condition × Migration (Nonmigrant) | −45.21 | −96.45 | 5.14 | 1980.000 | 0.086 |
| **Condition × Domestication Status** | 125.29 | 78.65 | 171.35 | 1980.000 | **<0.001** |

Individual ID was nested within the study site, and species and phylogeny were included in the model as random effects to control for nonphylogenetic and phylogenetic nonindependence, respectively, among individuals. $p_{MCMC}$ is twice the posterior probability that the estimate is negative or positive (whichever probability is smallest), L. CI = lower 95 credible interval, U. CI = upper 95 credible interval. Significant predictors ($p < 0.05$) are indicated in bold. Eff. samp is the effective sample size of a Monte Carlo computation.

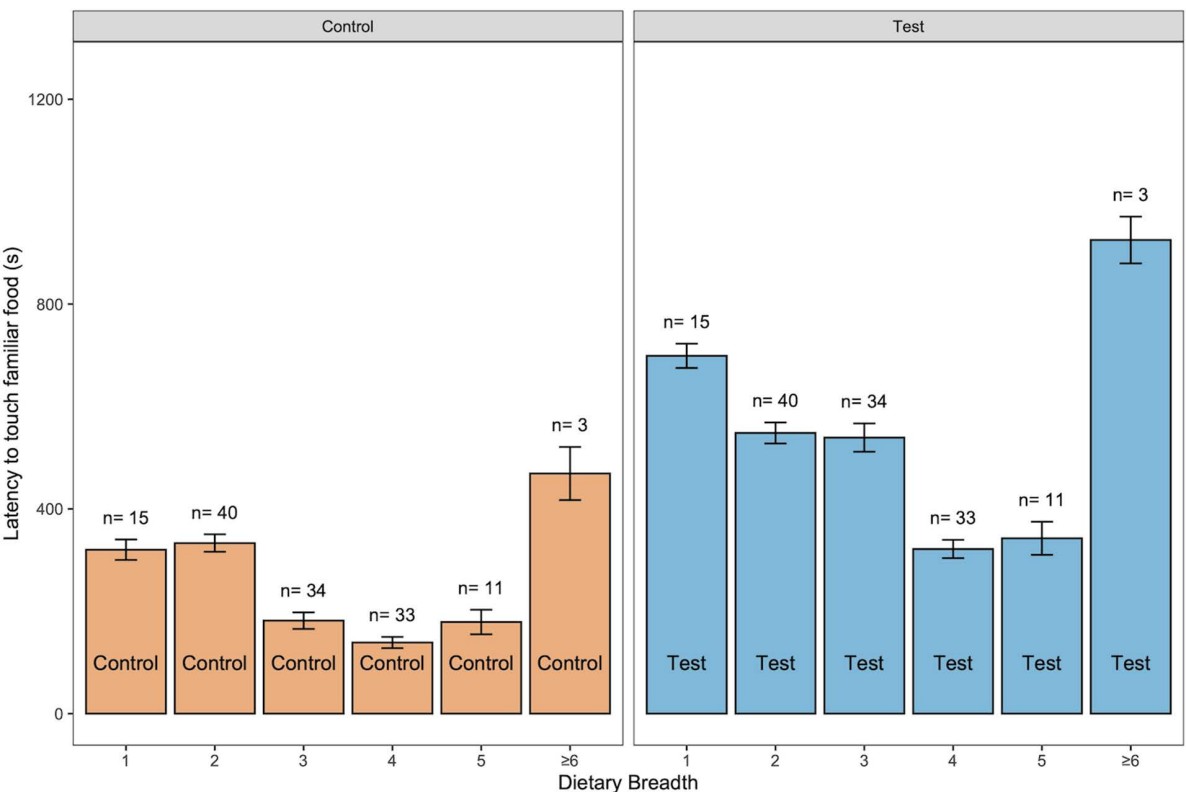

**Fig 4. Mean ± SE latency of birds to touch food in control (orange) and novel object (blue) conditions in relation to dietary breadth, the number of food categories consumed.** Species sample sizes for each dietary breadth category are given above the bars. The data underlying this figure can be found in https://doi.org/10.6084/m9.figshare.27324972.

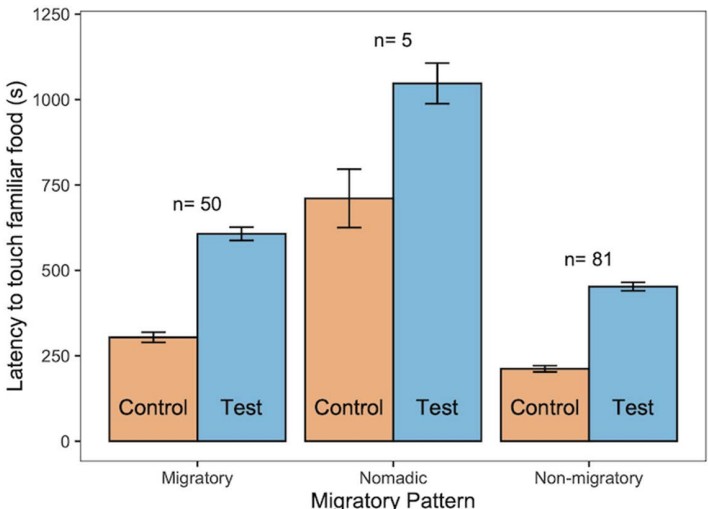

**Fig 5. Mean ± SE latency of birds to touch food in control (orange) and novel object test (blue) conditions, in relation to migratory pattern.** Species sample sizes for each type of migratory pattern are given above the bars. The data underlying this figure can be found in https://doi.org/10.6084/m9.figshare.27324972.

domesticated), showed higher object neophobia—i.e., they took longer to touch familiar food in the presence of novel objects (Table 3). Furthermore, individuals tested in a social context (versus alone) were more neophobic. Under control conditions, migratory (versus nonmigratory; Fig 5) species took longer to touch familiar food when a novel object was not present (Table 3). Trial order had an effect on neophobia with more neophobic responses for the trials that were presented second (they could be a control or a novel object test depending on the subject and the round), compared to responses obtained for trials presented first.

Focusing the analysis on a reduced data set of nondomesticated species (Model D) only (i.e., excluding seven domesticated lineages; raw data across species) did not change the findings on dietary breadth or migratory pattern (Table F in S1 Text). However, there were interacting effects of condition with habitat complexity, with territoriality, and with social context. Specifically, species from less complex habitats took longer to touch food in both control and novel object test conditions than those from more complex habitats (Table F in S1 Text). In the novel object test, species with year-round territorial systems were more neophobic than those with seasonal territoriality (Fig D in S1 Text) and those tested in a social context were more neophobic than those tested alone (Table F in S1 Text). Trial order had no effect.

## Discussion

In the largest multispecies standardized comparative study on neophobia, conducted through the ManyBirds Project, we aimed to establish whether neophobia is a stable trait within individuals and affected by phylogeny, and to identify the ecological drivers of neophobia across the avian clade. In line with our predictions, we found that object neophobia was (1) repeatable across time and contexts, and (2) related to phylogeny driven by ecological factors relating to diet, habitat, range, social context, and domestication status. Specifically, latency to touch familiar food in the presence of a novel object varied across the 25 avian orders represented in our sample. Across all 136 species tested, using difference scores (control latencies subtracted from novel object latencies), we found that bird species with a more restricted dietary breadth and migratory species were more object neophobic than species with a broader dietary breadth and nonmigratory species, respectively, consistent with predictions of the Neophobia Threshold Hypothesis (for diet) and Dangerous Niche Hypothesis (for migratory patterns). Using raw data, neophobia was more pronounced among species that were feeding and habitat specialists, migratory and nondomesticated than among species that were feeding and habitat generalists, nonmigratory and domesticated, which supported some of the predictions of all three theoretical frameworks: the Neophobia Threshold Hypothesis, Dangerous Niche Hypothesis, and Life History Theory. When we removed seven domesticated lineages from the sample, most of these findings also held for nondomesticated species, and social factors became more important. Specifically, species with year-round territoriality were more neophobic than seasonally territorial species (as predicted by the Neophobia Threshold Hypothesis) and species tested in a social setting showed higher neophobia than those tested alone. Finally, trial order affected neophobia only for the raw data and all species analysis - somewhat surprisingly, trials presented second elicited higher neophobia than those presented first.

## Repeatability

Individual responses to familiar food (presented alone) and to familiar food in the presence of novel objects were repeatable over test rounds. Furthermore, repeatability findings held at species, family, and order levels, although repeatability decreased at the higher taxonomic levels. These findings highlight another benefit of using a Big Team Science approach as few previous studies have been able to demonstrate and compare consistent responses within- and between-species. It also supports the use of captive birds, as neophobia responses are conserved. In previous studies, individuals either showed consistency or flexibility (e.g., shift in responses, inconsistency, or lack of repeatability) to novelty [15,67]. There may be some groups of birds where responses are less repeatable due to factors such as rapid learning, i.e., habituation to novelty, so studies focusing on species within those taxonomic groups may be less likely to find repeatability. However, taxonomic distribution in our data

limits testing this with our sample. Variation in neophobia repeatability may also be due to factors such as developmental and social influences [68–70]. Repeatability can also vary temporally. While many studies, including ours, find high repeatability across shorter timespans [56], this is not always the case; variation in results can depend on the behavior measured and the number and frequency of measures recorded within designated timeframes [71]. We used a consistent methodology across species, and while subjects varied in ontogenetic background (e.g., wild versus captive) and whether they were tested socially versus alone, these variables were held constant within individuals and sites. Our results suggest that neophobic responses reflect a stable behavioral trait that is consistently different among individuals, which may reflect underlying individual differences in neurobiology or endocrinology [72,73] and is a prerequisite for heritable variation. Thus, object neophobia appears to be consistent within individual birds (at least for the short time spans tested in our study), while it is less repeatable among species.

### Ecological drivers of neophobia

We found evidence in support of the Neophobia Threshold Hypothesis, including effects of dietary breadth and habitat use diversity [19,23], the Dangerous Niche Hypothesis, including an effect of migratory pattern [24], and Life History Theory, including an effect of domestication [29]. Previously, Miller and colleagues [15] identified four socio-ecological correlates of object neophobia in 10 corvid species: urban habitat use diversity, adult sociality (i.e., territorial versus family group living), maximum flock size, and caching behavior. However, this study did not find effects of geographic range, foraging differences, or genus. Although the specific factors examined differed between this and the corvid study, we similarly found effects related to sociality and habitat. However, because the tested corvid taxa have largely similar diets, degree of dietary specialization could not be examined relative to neophobia as it could in the present study. Furthermore, given the broader representation of species in this study, we were able to adequately explore phylogenetic effects in our analyses and identify evolutionary drivers of neophobia at a larger taxonomic scale.

### Diet

We expected feeding generalists to be less neophobic than specialists, as they are usually more likely to forage in unfamiliar situations and interact with a wider range of stimuli than specialists [19,74,75]. Indeed, previous studies reported that feeding generalists are typically faster to explore and exploit different food types [4,13,19,76]. Our findings, based on both raw data and difference scores (as well as across all species and for nondomesticated species only), support this prediction broadly across birds as specialists were found to exhibit higher levels of neophobia, and neophobia decreased with increasing generalism. Specialized species experience fewer variations in environmental stimuli and consequently may perceive changes as more threatening than generalists. We note that only three of the included bird species had very high dietary breadths, receiving classifications of 6 and 7 (red-crowned crane, *Grus japonensis*; greater rhea, *Rhea americana*; and European starling, *Sturnus vulgaris*) therefore, the trend was driven by the 133 species with dietary breadth 1–5. Indeed, two of these three species were highly neophobic, thus we would expect an opposite trend (or a u-shaped distribution) if these few values had a disproportionate impact on the coefficient.

### Territoriality

We expected territoriality to play a role, whereby less territorial species might be thought of as "spatial generalists" compared to more territorial "spatial specialists;" therefore, we predicted less territorial species would show less object neophobia as per the Neophobia Threshold Hypothesis. We found some evidence supporting this prediction (raw data for nondomesticated species only), tentatively suggesting that neophobia could act as a proximate mechanism for niche specialization in terms of territoriality as well as diet (or alternatively that niche specialization selects for greater neophobia, or both), an exciting extension of this hypothesis. Our findings are in line with data on corvids, suggesting that territoriality correlates positively with neophobia [15]. Age-specific analyses may help clarify this relationship further, since nonbreeders—such as juveniles and floaters—can exhibit more vagrant, nonterritorial behavior and form large, dynamic

flocks, potentially functioning as "spatial generalists." In contrast, paired territorial adults are more spatially fixed and may be more neophobic as "spatial specialists" [77]. Such developmental transitions in spatial behavior and social structure, particularly in long-term monogamous species, could therefore shape age-related neophobia profiles and refine our understanding of how territoriality and neophobia interact.

## Habitat and range

Migratory species were more neophobic than nonmigratory species in both primary analyses, fitting predictions of the Dangerous Niche Hypothesis. This finding may relate to increased risk associated with interacting with a wide range of potentially dangerous novel items and environments, or increased predation risk that migratory species may encounter. Increased neophobia may be beneficial in species that encounter more unfamiliar sites, as migratory species only stay for short periods in one particular area, while nonmigratory species are more familiar with changes and threats in their territorial range throughout the year [24,78]. These findings are in line with previous studies in closely related species; for instance, neophobia was higher in the migratory garden warbler (*Sylvia borin*) than in the resident Sardinian warbler (*Curruca melanocephala*) [78], and in migratory New World blackbirds (Icteridae) than in resident species of the same taxonomic family [24].

In addition, as predicted by the Neophobia Threshold Hypothesis, we expected generalist versus specialist habitat use (i.e., occupying many versus few habitat types) and habitat complexity to affect neophobia. Indeed, using raw data and across all species, species with less complex habitats and that used fewer habitat categories (i.e., more specialist species) were more neophobic than species with more complex habitats and that used more habitat categories (i.e., more generalist species). Anthropogenic habitat use was examined separately, as these habitats may be a special case of complex habitats that are more dynamically changing than most natural environments (see Table 1). In anthropogenically-modified environments, neophilia (i.e., attraction to novelty) may be more adaptive for survival. However, the association between neophilia and neophobia in urban environments has not been adequately investigated so far [79]. In our study, we found no effect of anthropogenically-modified habitat use on object neophobia (note that the defined 'anthropogenically modified habitat' is not synonymous with 'urban habitat use', see Table 1 definition). This finding did not appear to be due to a lack of representation of urban exploiters, which represented 63% of species in our sample. However, for pragmatic reasons, we classified anthropogenic habitat use in birds into only two categories (yes/no) which limited the resolution of our analysis. We also could not account for population of origin in our analysis (i.e., deriving from anthropogenically modified or unmodified habitats, as many of our subjects were captive-bred zoo animals descending from unknown founder populations). However, our results, limited as they are, provide no support for the view that a bird species' ability to inhabit anthropogenic habitats arises from it being "pre-adapted" to urban life due to low levels of neophobia [80].

## Other factors

We included domestication status and body mass as other factors that may influence neophobia. Note that body mass was one case where we had opposing predictions based on the Dangerous Niche Hypothesis (which would predict lower neophobia in larger birds, as predation is often lower on larger species; [62] and Life History Theory (which would predict more neophobia in larger birds, as larger-bodied species tend to have slower life histories and be more risk-averse; [81]). Body mass may also serve as a potential control for motivation as smaller species with higher metabolic rates may need to forage more frequently and thus have shorter latencies for control and novel object test trials. However, we did not find any effect of body mass on neophobia. We note that average mass may mask sexual dimorphism (where it exists) and age-related development, thus this factor is worth exploring further in future datasets that facilitate these comparisons. Domesticated species were less neophobic than nondomesticated species (raw data only), which was expected since reduced neophobia is often a

desired trait of domesticated lineages, or may result indirectly from selective breeding for reduced reactivity [65]. This result is also consistent with predictions of both the Dangerous Niche Hypothesis and Life History Theory.

We also tested the role of social context (whether birds were tested individually or in groups) in neophobia; based on hypotheses related to animal behavior, group size, and potential danger [82–85], we expected birds to show less neophobia in groups. However, we found the opposite result: birds tested in groups were more neophobic than birds tested alone. Similar patterns have been found in previous studies, e.g., in a within-subjects comparison, corvids took longer to interact (then interacted more frequently) with novel items when one or more conspecifics were present [33]. Similarly, house sparrows that had previously shown low object neophobia took longer to approach novelty when tested with neophobic, but not nonneophobic, conspecifics [17]. These results may reflect a "socially induced" neophobia, where individuals wait to allow others to take the risk of approaching first and/or approach behavior is affected by the fear or anxiety cues of more neophobic conspecifics [17,86]; alternatively, dominance relations may prevent quicker approaches to familiar food [87].

Trial order had an unexpected effect on the raw data (all species included), though no effect on the difference score data or on the raw data for nondomesticated species. While we did not expect trial order to influence behavior, if any such effect were to occur, one might predict higher neophobia in each individual's first trial, as it represents the initial exposure to the testing situation. However, we found the opposite pattern. Sensitization learning seems an unlikely explanation for this result, as the trials were few and well-spaced over time. Satiety or lower motivation could explain longer latencies, but we can rule out this explanation as the majority of trials occurred on different days (trials were separated by at least 24 h), and there was no effect of trial order for the few subjects with both trials on the same day. Therefore, this finding may potentially be an artifact relating to the slightly unbalanced data (i.e., 60% of first trials were control trials).

## Limitations and future directions

Large-scale collaborations provide multiple advantages [43,88] but also pose challenges, such as increasing the amount of uncontrollable variance in the dataset through differences in testing conditions based on site-specific factors (e.g., tested social versus alone which was included as a factor in our models; test arena size) and the prior history of study subjects (e.g., whether they were wild or captive-bred; subject age) [89]. For instance, although we were open to including neophobia data from wild animals, 90% of our subjects were sampled in lab and zoo captive settings rather than field settings, potentially reflecting logistic constraints in some cases (e.g., difficulties in identifying unique individuals in the field, obtaining permits for fieldwork, higher logistic costs, etc.). The large percentage of captive birds sampled can be viewed as a strength of the study, given that captive settings are more controlled and more easily standardized. While it is possible that individual differences in neophobic responses are less pronounced in captivity, neophobia has been shown in songbirds to be consistent between natural and captive settings [90].

Our study sample reflects a general over-representation of captive sites in avian cognition research but diverges from previous work by including a large number of zoo-housed subjects (see [41]). It may indicate a form of 'self-selection' by contributors, such that those with access to captive animals were more willing or able to collect the relevant data within the specified time frame, for example, through short-term student projects. We controlled for this variance wherever possible in the analyses, by including factors like domestication status and by analyzing difference scores as well as raw data. We also provided as many details as possible for each study site (Table G in S1 Text), allowing for possible future investigations into the effects of site-specific factors not considered in the present analyses.

Nonetheless, in the present study, it was not possible to directly test the effects of captivity on object neophobia due to the relatively small sample of wild species tested (which were also taxonomically skewed). Available evidence suggests that captive bird populations may differ drastically from wild ones regarding various aspects of behavior and cognition [91], placing an important caveat on studies that, like ours, rely heavily on captive-bred subjects from zoos and laboratories. There is also data suggesting marked differences between species regarding chronic stress in captive settings, which

might be pronounced (even if the individuals in question have been raised in and used to captivity) and can be influenced by inconspicuous differences in housing conditions [92–94]. Our results should therefore be considered with these limitations in mind, and we encourage more comparative work explicitly testing effects of captivity on neophobia.

Another potential limitation of our study relates to the standardization of the novel object stimuli across the broad taxonomic range of species tested. While every effort was made to ensure consistency in object construction and presentation (e.g., by using a standardized color palette, shiny texture in part of the object, and scaled size), species may nonetheless differ in how they perceive or interpret specific object features due to species differences in sensory physiology or ecology. For example, frugivorous species might be more accustomed to or attracted by brightly colored objects, potentially resulting in lower observed neophobia that may not reflect broader novelty responses [95]. While such perceptual or ecological differences are difficult to eliminate entirely in large comparative datasets, future studies should build on this work by testing responses to different object types that vary in specific features (e.g., color, texture, shape) to further disentangle perceptual biases from general neophobic tendencies.

Regardless of the inclusive approach taken to sample selection and authorship, some avian taxa were substantially more represented than others, due to increased access to those species rather than *a priori* species selection. Therefore, some factor coding categories were somewhat unbalanced, such as migratory habits, where nomadic birds are represented by five species only. We remain conservative in our interpretation in these instances, and care should be taken with wider generalization of these findings. Furthermore, the variation in the availability of socioecological and life history trait information across species due to skewed species representation in the avian literature [96] may have introduced noise into our ecological coding and could obscure real effects.

Even with a clear and relatively simple protocol, ensuring that all data were collected in as comparable a manner as possible was a challenge. All collaborators were required to submit pilot videos of their testing conditions and photograph examples of their novel objects for approval by the leadership team prior to data collection (e.g., to check camera angles and positioning of familiar foods and objects). Novel objects were required to follow the same approximate principles regarding colors and textures and were adjusted for species size. In field or some zoo settings, it was difficult to identify individuals; in these cases, individual data points were not included in the repeatability analysis or were only tested in one round. As it has been found that seasonality can influence neophobia in some species [97], other limitations of this study include variation in testing across different seasons or breeding stages of subjects.

To facilitate standardization by multiple researchers across many species and sites, we focused on object neophobia only. Future studies may expand to other types of neophobia, such as novel foods, predators, and environments [97,98]. We have focused primarily on ultimate explanations for neophobia [99] but there are many potential proximate drivers of neophobia that were not possible to assess with this study and sample, such as age effects [100], whereby younger birds may be less neophobic than adults [33,101], and hormonal effects, whereby higher levels of prenatal testosterone and lower levels of adult corticosterone are often linked to lower neophobia [97,102–104]. Future studies could also further examine the relation between neophobia and other cognitive and behavioral measures, such as problem-solving and innovation [21].

Body mass and domestication status were the only life history-related predictors coded and tested with our full dataset, with no effect found for body mass and domesticated birds showing lower neophobia, as predicted by Life History Theory and previous work on domesticated birds [29]. Although additional variables (e.g., clutch size) could have been added to this analysis, there was a trade-off between maximizing the value of this particular dataset and the risk of finding false positives from running many analyses, especially considering that we were already examining raw data and differences score data for each variable, both with and without domesticated species. Future studies on object neophobia should focus more on the influence of life history factors such as longevity, developmental speed, and reproductive strategy (r/K strategies). Furthermore, potential effects of neurological variables may be explored, which we did not consider here due to the ecological focus of this study. Future ManyBirds projects may include more targeted species selection, as well as

increasing the number of individuals per species as appropriate for the specific research question, with a planned consensus-based approach for study selection.

## Conclusions

Our understanding of the evolutionary drivers of neophobia has been limited by a lack of taxonomically diverse comparative studies. Drawing on a sample of 136 bird species, we examined the repeatability of object neophobia at multiple taxonomic levels and tested predictions regarding eight ecological variables in relation to interspecific neophobic variance. Across four models drawing on both raw data and difference scores as well as data subset for nondomesticated species, our results revealed robust support for predictions of the Neophobia Threshold Hypothesis and Dangerous Niche Hypothesis, in that both dietary breadth and migratory habits were, as expected, predictors of neophobia. Specifically, feeding generalists were less neophobic than feeding specialists and migratory species were more neophobic than non-migratory species. We also found some evidence for additional effects of territoriality, habitat complexity and diversity, and domestication impacting neophobic responses. Namely, less territorial species showed lower object neophobia than more territorial species, species with less complex habitats and those species considered as more habitat specialists were more neophobic than species with more complex habitats and those considered as more habitat generalists, and domesticated species were less neophobic than nondomesticated species. Why, then, do dietary breadth and migratory habits stand out as robust predictors of neophobic behavior? Different selection pressures are likely to affect each, making them subject to two distinct hypotheses (the Dangerous Niche Hypothesis for migration, and the Neophobia Threshold Hypothesis for feeding preferences). Our results suggest that neophobia is under selection pressure in a migration context because of risk mitigation, but is less important in other contexts such as navigating social situations. For feeding behavior, neophobia may act more as a constraint, restricting the evolution of a generalist feeding strategy, and this may be stronger for feeding than for traits such as territoriality or habitat complexity. Our findings make clear advances in our understanding of the ecological drivers of neophobia, while providing support for two of the central hypotheses explaining the evolution of this behavior. This study, therefore, lays a strong foundation for future neophobia studies to build from.

## Materials and methods

### Subjects, species, and data collection sites

This study aimed to include data from as many bird species available as possible, across labs, zoos, and field sites with an open call for data contributions from January 2022 to April 2023. This study included 1,439 subjects and 136 species, across 25 taxonomic orders (Figs 1 and 2 and Table A in S1 Text). The taxonomy and nomenclature of the Clements checklist of Birds of the World [105] was used. We included as many subjects as possible, with no limitations on bird sex or age. It was useful, though not essential, for individuals to be identifiable (e.g., via leg rings, wing bands, cable ties, plumage coloration, or body size differences) and to be able to be tested while alone (e.g., temporary visual isolation). The majority of data collected constitutes new data, i.e., it was collected for this study. Some data were collected using the same protocol and published separately. Specifically, we included a dataset on neophobia in five hornbill species (family Bucerotidae) [46] and in Bali myna (*Leucopsar rothschildi)* [106] with permission or collaboration with relevant authors. Finally, we included some additional published and new data (dataset available via Figshare https://doi.org/10.6084/m9.figshare.27324972), but excluded these data from analysis due to having a maximum trial length of 600 s per trial, shorter than other trials (previously published data on 10 corvid species, [15], as well as some new data on four species—rock dove, *Columba livia,* Japanese quail, *Coturnix japonica,* budgerigar, *Melopsittacus undulatus,* and zebra finch, *Taeniopygia guttata*). Note that new data from Japanese quails, budgerigars, and zebra finches (where max trial length was over 600 s) were included in analysis.

## Materials

There were two experimental conditions: control (familiar food alone) and novel object test (familiar food placed beside the novel object). The familiar food, placed in a familiar food bowl, was species-appropriate and varied between bird groups, depending on the regular diet from each site. There were one to three types of novel objects (depending on number of test rounds conducted per site and species) with the same visual novelty properties (i.e., not auditory or olfactory novelty). Novel objects were made of multiple items and textures, with no part that could look like eyes (to avoid resembling predators), and all contained the colors blue, yellow, green, and red [67]. Parts of the objects were also shiny. The objects were between one-third and one-half of the subject's size, so that the object size varied with species (Fig 6). Objects were constructed by the primary collaborator with a photograph shared for checking and approval by the ManyBirds study 1 team prior to data collection commencement. Where possible, the birds were tested in a familiar area (e.g., the regular feeding location) or following necessary habituation prior to testing. The test sites gave the birds as much room as possible to avoid or approach stimuli. In a few rare cases, for five zoo-housed species at Basel Zoo (kea, *Nestor notabilis;* gentoo penguin, *Pygoscelis papua*; king penguin, *Aptenodytes patagonicus*; African penguin, *Spheniscus demersus*; Southern ground-hornbill, *Bucorvus leadbeateri*), the novel object was placed in a transparent box to prevent the birds

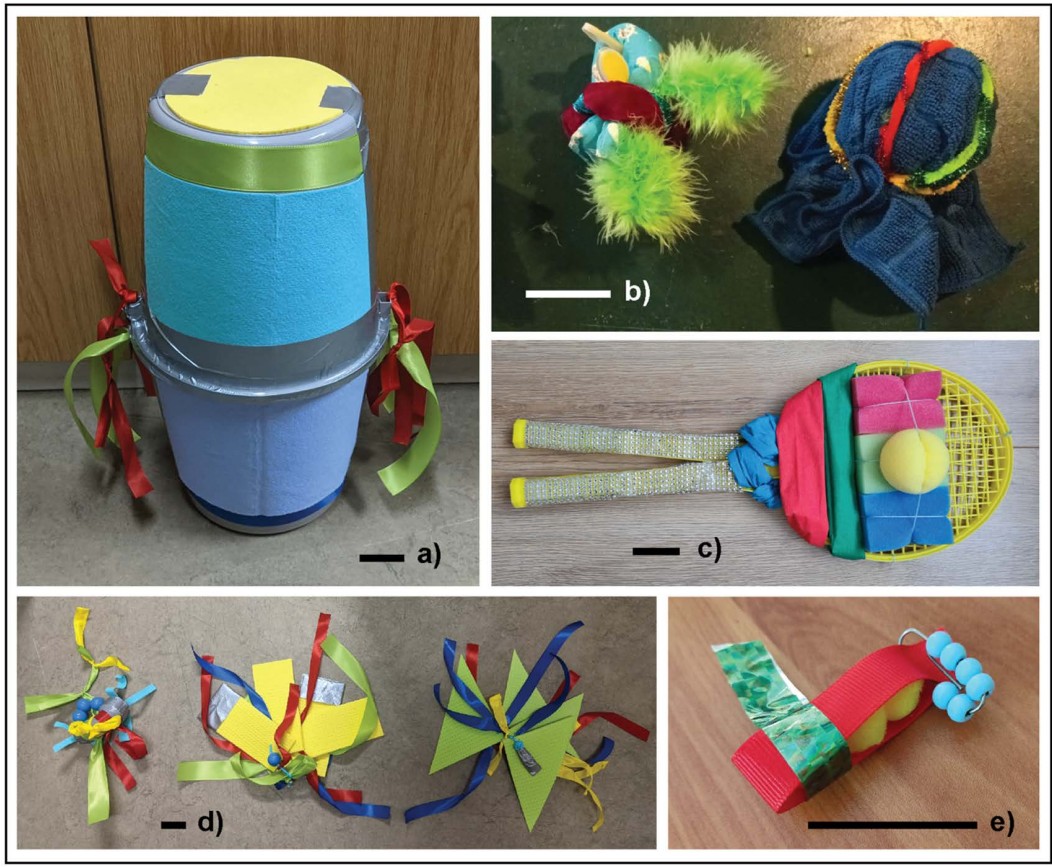

**Fig 6. Examples of novel objects used.** Each black/white bar is 5 cm long. The objects ranged in size from a third to half the size of the subjects: **(a)** Southern cassowary, *Casuarius casuarius*, **(b)** Moluccan eclectus, *Eclectus roratus*, **(c)** Rüppell's vulture, *Gyps rueppelli*, **(d)** gray-winged trumpeter, *Psophia crepitans*, **(e)** common waxbill, *Estrilda astrild.*

from touching it, a zoo-specific requirement. These birds were thus given additional trials presenting the empty box without a novel object beside food to habituate them to the presence of the box.

## Procedure

The novel object tests involved measuring behavioral responses to novel objects placed beside familiar food. The controls involved measuring behavioral responses to familiar food only. Testing occurred either (a) in the presence or (b) absence (i.e., temporary visual separation or isolation) of conspecifics, with 56% of subjects tested socially (i.e., conspecifics/heterospecifics present, including field sites where individuals could not be isolated for testing) and 44% of subjects tested alone. For captive subjects, presence or absence of others was consistent within individuals and sites. Where separation was not possible, the number of test stimuli (i.e., novel objects present per trial) and familiar food items were increased, respectively, to the number of subjects. For example, if two subjects were present, two separate test sites were offered, each site with a familiar food bowl and a novel object. Although species were not typically food-deprived in a standardized manner (because this was not possible for subjects in the field sites), testing took place either in the morning or alongside the usual main daily feeding (i.e., first or main feed of the day for captive subjects) to ensure that birds showed similar hunger and motivation levels for obtaining food.

The novel object was placed beside the familiar food-filled dish (distance: ~20 cm for larger species, such as Madagascar ibis, *Lophotibis cristata*; ~10 cm for smaller species, such as smaller passerines), with items placed in the same location (e.g., a table/platform/mesh wall—large enough so that the bird could approach it slowly from more than a body length away) for all tests and individuals within each species. The stimuli were either presented before the subject entered the testing area or introduced together with the food while the subjects were already present to accommodate variation in site logistics (e.g., depending on whether subjects were already present in the testing area or not per standard testing procedures at each site). The trial started when the subject entered the testing area or the experimenter left the test area following standard testing protocols for each site (e.g., the experimenter remained visible to subjects if this was standard protocol, thus these subjects were familiar with this human presence, or the experimenter left, and behavior was recorded using video cameras otherwise). Trial length was determined by a pilot control trial with familiar food only to check when each individual/species would usually approach during their daily feed. This was expected to be a maximum of 20 min based on previous research (e.g., [106]). Each trial ended after the maximum trial length in order to code all variables and all individuals, but only if it was appropriate for the given site or species tested. For example, if an individual was temporarily separated from conspecifics for testing, the full trial length only ran provided the individual was not stressed by this extended separation time (per ethical approval considerations). Whenever it was not possible to reach the maximum trial length, trials ended when the subject touched the familiar food (i.e., with beak or body) (this occurred at 24 sites; Table G in S1 Text). As we used latency to touch familiar food (i.e., first touch) as the primary variable, any influence of trial length differences between sites on results should be minimal.

Each test round consisted of one novel object test and one control, a total of two trials. Every test round was conducted between one and three times (1 trial per condition per round, 6 trials in total) - the latter allowed for individual repeatability testing. For example, only one test round was conducted in cases where individuals were not individually identifiable (e.g., some zoo and field sites; age and sex were thus unknown for these individuals, given they could not be tracked reliably across trials), as the individual-level responses could not be accurately compared between rounds. The control trial was conducted within 48 h of the novel object test trial to ensure that hunger/motivation levels were as comparable as possible between the conditions while also accommodating variation in site logistics (e.g., in feeding schedules and subject availability, as some species only fed once per day). Wherever possible, the control trial was conducted on a separate day (except for 36 individuals of 9 species), rather than immediately before or after the novel object test trial, to avoid potential carryover effects such as increased motivation, inhibition or arousal. Eighty-five individuals of 23 species were tested with a longer interval between trials (median: 3 days) due to constraints in

the operations of the host facility. However, during this interval, no changes in housing conditions or diet were made to ensure that hunger and motivation levels were as comparable as possible between control and novel object test conditions. Test rounds were repeated after approximately 2 weeks. Therefore, testing took between 1 and 6–8 weeks to complete per species and site.

At sites where more than one type of novel object was created for testing (i.e., more than 1 test round was run), the same type of novel object was used for test round 1 across all individuals, to ensure that test round 1 was comparable between sites. Exceptions were tests of blue-faced honeyeaters (*Entomyzon cyanotis*; *n* = 5), Fischer's turaco (*Tauraco fischeri*; *n* = 3), Marabou storks (*Leptoptilos crumenifer; n* = 2), Southern cassowaries (*Casuarius casuarius; n* = 2), and Visayan hornbills (*Penelopides panini*, *n* = 2) at Zoo Frankfurt, where the presentation order of novel objects was fully randomized. The order of presentation of the novel objects as well as the object types in test round 2 and 3 was, however, counterbalanced across subjects and species, e.g., subject 1, round 1—novel object type 1, round 2—type 2, round 3—type 3; subject 2, round 1—type 1, round 2—type 3, round 3—type 2, etc. The testing schedule for half of the subjects was control-object in every round, and for the other half object-control in every round per group (Table J in S1 Text). We included "test order" (difference scores) or "trial order" (raw data) as a fixed effect in the models.

Our main measure assessed was latency to touch familiar food, which signifies how long the subject took to touch a familiar, desirable food in the presence of a novel object (during novel object test trials) or without novel objects (during control trials). Any avoidance of the novel object (and thus familiar food) was then interpreted as neophobia [2]. We used latency to touch familiar food rather than latency to eat familiar food to control for any potential bias in terms of whether the bird swallowed the food. Example video trials can be found at: https://youtu.be/xGPQ6lcRGpE. Pre-registration with protocol and analysis plan was published in January 2022: https://osf.io/vdbks/?view_only=3d1a68898a0145c0b0f9c5a1b7b30333.

## Data analysis

### Ecological and other factor coding

As outlined in Table 1 (Data Source column) and via the full species list (ecological coding in Table A in S1 Text), ecological and other factors were coded for each species using or informed by peer-reviewed or reputable sources, where available. The selection of these factors was determined by the final species representation in the dataset, the reliability of coding each factor (e.g., available peer-reviewed sources) and was hypothesis driven (Table 1).

### Filming and coding

All trials were video recorded with high-resolution cameras. Camera position was the same across trials, at least within the same subject. The familiar food and, in the test trials, the novel object were visible on screen at all times. Recordings were coded frame by frame, but at least with a sampling frequency of 1.0 frame per second. Coding software included Microsoft Excel, Solomon Coder (https://solomoncoder.com) or BORIS [107]. Prior to data collection, each primary collaborator submitted a pilot video (familiar food only) to the Study 1 leadership team for approval. The start of the trial was clearly denotable and, although it varied between different sites depending on species and setup, was consistent between trials. For instance, this was either when (i) the animal entered the testing area by itself, (ii) the animal was released into a testing cage, or (iii) the experimenter left the test area. Under the responsibility of each primary collaborator, 12% of video trials were coded, for each species and enclosure, by a second coder and checked to ensure significant and sufficiently strong inter-rater reliability (i.e., min of 0.8) prior to coding the remaining data (Table B in S1 Text).

We considered both individual birds and "flocks" for the analyses and included the social context in which individuals were tested (alone/social) as a fixed effect in the models. Flocks were only coded when the individual identification of a bird within a group was ambiguous. In that case, the first interaction of any individual within the flock with the familiar food

(or novel object) was coded. Hence, latency to approach/touch food/novel objects were not necessarily determined from the same individual across different trials. For the analyses, flock-related data were dealt with analogously to those from individual birds. Thus, data from a flock were weighted as if they corresponded to a single bird. In total, flocks were coded for nine species. In five of those species only, all data came from flocks (speckled mousebird, *Colius striatus*; Griffon vulture, *Gyps fulvus*; carmine bee-eater, *Merops nubicus*; black-crowned night heron, *Nycticorax nycticorax*; blue-naped mousebird, *Urocolius macrourus*). In the other four species (Cinereous vulture, *Aegypius monachus*; white stork, *Ciconia ciconia*; white-faced whistling duck, *Dendrocygna viduata*; Inca tern, *Larosterna inca*), data from flocks and individual birds (i.e., where it was possible to identify individuals consistently) were pooled.

## Statistical analysis

For all between-species and order comparisons, we tested the full dataset using raw data (control and novel object test latencies separately) and difference scores with the same model parameters and both fixed and random effects, with interaction with neophobia conditions (control/novel object test) (Objective 2 below). Difference scores, i.e., latencies of control responses (only familiar food) subtracted from the latencies of novel object tests aimed to help in standardizing latencies across sites as well as control for baseline neophobia and current motivational state [2,15]. Raw data analyses tested for effects within each condition, i.e., within novel object and control trials, while difference scores used a composite score across conditions. For within-species comparisons at the individual level (Objective 1 below), both raw data and difference scores were used (as per [15]). For each objective, we carried out analyses in R version 4.3.1 [108]. Given the nature of the dataset, e.g., testing multiple individuals per species at multiple sites and phylogenetic dependence, we used Bayesian mixed models (Objective 2). In the following paragraphs, we outline the statistical analysis.

Objective 1 was to test for the extent to which variance in the data was explained by consistent differences among individuals, among species, among families, and among orders, temporal (i.e., test round) and contextual (i.e., novel object test and control condition) levels. We extracted "R" (repeatability) estimates from models with individual as a random effect and ran a bootstrap (e.g., 1,000 samples) to generate 95% confidence intervals around the estimates (R package rptR [109], using rpt() function). For contextual repeatability, we included "condition" (control, novel object test) in the model and for temporal repeatability, we included "round", with "individual ID" fitted as a random effect for individual repeatability, or nested random effect for species, family, or order repeatability (as per [15]). We did not include sex or age as effects as we did not have this information for all subjects (e.g., zoo/field sites). We compared individual repeatability within each condition (i.e., control and novel object test conditions separately) and using latency difference scores. Finally, we tested repeatability with body mass, dietary breadth, habitat complexity, anthropogenic habitat use, habitat use diversity, territoriality, migratory pattern, and domestication status as fixed effects in the model.

Objective 2 was an avian order comparison of neophobia: that is, we tested whether taxonomic "order" was a predictor of the obtained data (all trials and test rounds available). The response variable was "latency to touch familiar food", with the fixed effect of taxonomic "order" and the random effect of "individual". Due to the large number of species represented ($N = 136$), we selected taxonomic order, rather than species (as listed in the pre-registration) for Objective 2, to facilitate capture of broad-scale differences and aid in interpretation of results. We used a Bayesian mixed model and analyzed each condition (control or novel object test) and the individual difference between control and novel object test conditions in each trial separately.

We then examined the effects of ecological and other factors on neophobia across species. The final selection of factors was dependent on the species representation in the final sample and driven by our hypotheses (Table 1; body mass—continuous, dietary breadth—continuous, habitat complexity—continuous, anthropogenic habitat use—categorical, habitat use diversity—continuous, territoriality—categorical, migratory pattern—categorical, domestication status—categorical). The response variables were the "latency to touch familiar food" and the latency difference scores, the main

effects were aforementioned factors, as well as "social context" (alone, social testing) and "test order" (when the *novel object test* was conducted; control-novel object test, novel object test-control, both same day) in the difference scores analysis or "trial order" (when the *trial* was conducted; first day—can be either novel object test or control trial, second day, both same day) in the raw latency data analysis. For the raw data analysis, we also included each main effect in inter-action with condition (control, novel object test). The random effects included were: "species", with "individual ID" nested within "study site", and "phylogeny" to control for nonphylogenetic and phylogenetic nonindependence, respectively, among individuals.

Models were run using Markov chain Monte Carlo (MCMC GLMM) estimation methods implemented in the MCMC-glmm package [110]. We applied an uninformative prior distribution ($V = 1$, nu $= 0.002$), and models were run for 100,000 iterations, 1,000 burn-in and a chain thinning of 50. We tested (1) the full dataset and (2) a reduced dataset of nondomesticated species only (removing the seven domesticated species), as we expected and found domestication status to have an effect on neophobia. Domestication was an additional included factor that was not a straightforward ecological variable (although they have ecological implications), as domesticated species traits are derived from their wild counterparts, thus domesticated species' data may influence the other variables. Further, habitat variables for domesticated species were coded in line with their wild counterparts, as they no longer have truly natural habitats, so differed to coding of nondomesticated species in this aspect.

For both models, we generated a single consensus phylogenetic tree from birdtree.org [54]; while more updated bird phylogenies exist (e.g., [111]), none of them has yet presented a species-level resolution that can be used with the species in our dataset, causing an issue for the analyses. While differences between newer bird trees and birdtree.org are small, the placement of some orders (most notably Musophagiformes, turacos) do differ; however, the number of potentially ambiguous species ($N = 3$) was small, and our findings were robust to these differences in tree structure. The final data set had 13 species tested in the field and 123 species tested in captivity (lab, zoo, temporary captivity). With the limited number of species tested in the field, we did not include "site type" (field/captivity) in the analysis, though we did include "site" (i.e., institution/organization name). A simple "wild" versus "captive" category was also not possible to reliably code, as some 'captive' subjects were wild-caught, first or second generation wild-caught, or of unknown origin. Similarly, for field and some zoo sites, sex ($N = 544$), age ($N = 322$), and rearing history ($N = 694$) were unknown or unreliable, there-fore, these factors were not included in the models. In sum, 65% of subjects touched the familiar food in the presence of a novel object in round 1 (75% in round 2). In round 1, 1,439 subjects were tested; in round 2, 639 subjects were tested; in round 3, 326 subjects were tested.

For both models (raw and difference score data), to ensure robust convergence diagnostics, we ran three separate chains of the model with different seeds. We assessed the convergence of the MCMC chains using the Gelman–Rubin diagnostic from the coda package [112]. The chains were combined into an mcmc.list object and the Gelman–Rubin diag-nostic was calculated, which indicated good convergence for all parameters in the model. For both models, the potential scale reduction factors were all close to 1, with upper confidence intervals also near to 1, suggesting that the chains have likely converged well, providing reliable estimates for the model parameters.

We presented results of both (1) difference scores and (2) raw data and found consistent effects of dietary breadth and migration pattern across both approaches. The difference scores are independent of the control latency and comparable between individuals and species. For example, two birds (A, B) may have a similar difference score (e.g., low neophobia), though bird B took longer to approach the familiar food than bird A (longer control latency), reflecting an individual/species difference in general approach to food (which may be influenced by, e.g., hunger, motivation, competition). It does not (necessarily) mean bird B is more neophobic than bird A, therefore, both raw and difference score data is informative for the magnitude of difference as well as where this occurs. For example, with the dietary breadth result, both control and test latencies decline at greater breadth, but the declines are greater under test conditions. Furthermore, from an open science perspective, we felt it was important to present both sets of findings for full transparency and highlight that the

selection of different analyses, or the inclusion or removal of certain species, like removing domesticated species, can result in different outcomes.

## Ethics statement

It was the responsibility of each primary collaborator to ensure that the appropriate ethical approval was obtained prior to data collection in relation to this study. Furthermore, we had an overarching ethical approval for the wider study protocol under a University of Cambridge nonregulated procedure (NR2023/12) to Rachael Miller. Ethics approval for each site is outlined in Table G in S1 Text.

## Supporting information

**S1 Text. Supporting Information. Table A:** Ecological factor coding. **Table B:** Inter-rater reliability. **Table C:** Repeatability (±95%CI) of birds' responses at different grouping levels across control, novel object test and difference scores. **Table D (a–c):** Generalized linear mixed models using Markov chain Monte Carlo estimation methods (MCMCglmm models), testing the effect of avian order on the latency to touch familiar food when no novel object was present (i.e., raw data) on (a) control, (b) novel object test conditions or the (c) difference scores (novel object minus control values of latency to touch familiar food). **Table E:** Generalized linear mixed models using Markov chain Monte Carlo estimation methods (MCMCglmm models), testing the effect of predictors on birds' difference scores in nondomesticated species only (seven domesticated species removed from data set; Model B). **Table F:** Generalized linear mixed models using Markov chain Monte Carlo techniques (MCMCglmm models), testing the effect of predictors on the difference (novel object minus control values) between control and novel object test latencies to touch familiar food (Model D). **Table G:** Site descriptions. **Table H:** Acknowledgements. **Table I:** Author contributions. **Table J:** Example test schedule and object ("obj") counterbalancing across subjects. **Fig A (a–c):** The mean ± SE latency of birds to touch food in (a) control and (b) novel object test conditions, for each avian order. The data underlying this figure can be found in https://doi.org/10.6084/m9.figshare.27324972. **Fig B:** Mean+SE latency to touch food in control (orange) and novel object test (blue) conditions, in relation to habitat complexity (1 = high density; 3 = low density). The data underlying this figure can be found in https://doi.org/10.6084/m9.figshare.27324972. **Fig C:** Mean ± SE latency to touch food in control (orange) and novel object test (blue) conditions in relation to habitat use diversity, the number of habitat categories where a species was found (i.e., more habitat generalists vs. specialists). The data underlying this figure can be found in https://doi.org/10.6084/m9.figshare.27324972. **Fig D:** Mean ± SE latency to touch food in control (orange) and novel object test (blue) conditions in territorial systems (nonterritorial, seasonal or year-round territories) in nondomesticated species only. The data underlying this figure can be found in https://doi.org/10.6084/m9.figshare.27324972. **Information A:** Avian order comparisons.
(DOCX)

## Acknowledgments

Thank you to all site managers, where appropriate, for providing access to birds for data collection purposes. Please see Table H in S1 Text for a full list.

**Contributions:**

Please see Table I in S1 Text for a full author contributions list. There were: 54 primary collaborators (i.e., someone who is "in charge" of a species/site data collection and so takes the main responsibility for this data contribution to MB1, e.g., a lab leader and/or supervisor); 75 contributors (i.e., usually someone who is funded and/or directly supervised in their MB1 contribution by a primary collaborator), and 129 authors in total. Authorship was determined following the ManyBirds

Project guidelines: https://themanybirds.com/authorship-guidelines-and-ethics-guidelines, which follow an adapted CRediT taxonomy (https://casrai.org/credit/).

## Author contributions

**Conceptualization:** ManyBirds Project, Rachael Miller, Vedrana Šlipogor, Stephan A. Reber, Megan Lambert.

**Data curation:** Rachael Miller, Vedrana Šlipogor, Kai R. Caspar, Jimena Lois-Milevicich, Carl D. Soulsbury, Stephan A. Reber, Megan Lambert.

**Formal analysis:** ManyBirds Project, Rachael Miller, Vedrana Šlipogor, Kai R. Caspar, Carl D. Soulsbury, Megan Lambert.

**Funding acquisition:** Rachael Miller.

**Investigation:** ManyBirds Project, Rachael Miller, Kai R. Caspar, Jimena Lois-Milevicich, Stephan A. Reber, Claudia Mettke-Hofmann, Megan Lambert, Benjamin J. Ashton, Melissa Bateson, Solenne Belle, Boris Bilčík, Laura M. Biondi, Francesco Bonadonna, Desiree Brucks, Michael W. Butler, Samuel P. Caro, Marion Charrier, Tiffany Chatelin, Johnathan Ching, Nicola S. Clayton, Benjamin J. Cluver, Ella B. Cochran, Francesca M. Cornero, Emily Danby, Samara Danel, Martina Darwich, James R. Davies, Alicia de la Colina, Dominik Fischer, Ondřej Fišer, Florencia Foitzick, Edward C. Galluccio, Clara Garcia-Co, Elias Garcia-Pelegrin, Isabelle George, Kai-Philipp Gladow, Anna Grewer, Katie Grice, Lauren M. Guilette, Devon C. Hallihan, Katie J. Harrington, Frauke Heer, Chloe Henry, Vladimira Hodova, Marisa Hoeschele, Cécilia Houdelier, Paula Ibáñez de Aldecoa, Yuka Kanemitsu, Mina Khodadadi, Duc Khong, Melanie G. Kimball, Ariana N. Klappert, Lucy N. Koch, Anastasia Krasheninnikova, Lubica Kubikova, Connor T. Lambert, Daan W. Laméris, Courtenay G. Lampert, Oceane Larousse, Christine R. Lattin, Michael Lindenmeier, Julia A. Mackenzie, Selina Mainz, Danna Masri, Jorg J. M. Massen, Laurenz Mohr, Paul M. Nealen, Andreas Nieder, Aurèle Novac, Nínive Paes Cavalcante, Kristina Pascual, Carla Pascual-Guàrdia, Ayushi Patel, Katarína Pichová, Laurent Prétôt, John Quinn, Elena Račevska, Sam Reynolds, Amanda R. Ridley, Theresa Rössler, Francisco Ruiz-Raya, Marina Salas, Beatriz C. Saldanha, Sebastián M. Santiago, Nikola Schlöglová, Gia Seatriz, Eva Serrano-Davies, Eva G. Shair Ali, Janja Sirovnik, Zuzana Skalná, Katie E. Slocombe, Masayo Soma, Tiziana Srdoc, Stefan Stanescu, Michaela Syrová, Alex H. Taylor, Christopher N. Templeton, Karlie Thompson, Sandra Trigo, Camille A. Troisi, Utku Urhan, Maurice Valbert, Alberto Velando, Jorrit W. Verkleij, Alizée Vernouillet, Jonas Verspeek, Petr Veselý, Eline Waalders, Benjamin A. Whittaker, Ella R. Williamson, Vanessa A. D. Wilson, Michelle A. Winfield, Neslihan Wittek, Karen K. L. Yeung, Jade A. Zanutto.

**Methodology:** ManyBirds Project, Rachael Miller, Vedrana Šlipogor, Stephan A. Reber, Claudia Mettke-Hofmann, Megan Lambert.

**Project administration:** ManyBirds Project, Rachael Miller, Vedrana Šlipogor, Kai R. Caspar, Jimena Lois-Milevicich, Stephan A. Reber, Megan Lambert.

**Resources:** Rachael Miller, Kai R. Caspar, Jimena Lois-Milevicich, Megan Lambert, Alice M. I. Auersperg, Boris Bilčík, Laura M. Biondi, Francesco Bonadonna, Michael W. Butler, Nicola S. Clayton, Alicia de la Colina, Isabelle George, Kai-Philipp Gladow, Anna Grewer, Lauren M. Guilette, Katie J. Harrington, Marisa Hoeschele, Paula Ibáñez de Aldecoa, Ariana N. Klappert, Uta U. König von Borstel, Ľubor Košťál, Lubica Kubikova, Daan W. Laméris, Christine R. Lattin, Zhongqiu Li, Michael Lindenmeier, Delia A. Lister, Jorg J. M. Massen, Wendt Müller, Paul M. Nealen, Andreas Nieder, Aurèle Novac, Katarína Pichová, Cristina Pilenga, John Quinn, Juan C. Reboreda, Francisco Ruiz-Raya, Marina Salas, Beatriz C. Saldanha, Eva Serrano-Davies, Janja Sirovnik, Masayo Soma, Michaela Syrová, Christopher N. Templeton, Sandra Trigo, Camille A. Troisi, Utku Urhan, Kees van Oers, Alberto Velando, Frederick Verbruggen, Alizée Vernouillet, Jonas Verspeek, Petr Veselý, Auguste M. P. von Bayern, Vanessa A. D. Wilson.

**Supervision:** ManyBirds Project, Rachael Miller, Kai R. Caspar, Jimena Lois-Milevicich, Claudia Mettke-Hofmann, Megan Lambert, Benjamin J. Ashton, Melissa Bateson, Michael W. Butler, Alicia de la Colina, Isabelle George, Lauren M. Guilette, Marisa Hoeschele, Paula Ibáñez de Aldecoa, Melanie G. Kimball, Uta U. König von Borstel, Ľubor Košťál, Anastasia Krasheninnikova, Christine R. Lattin, Zhongqiu Li, Jorg J. M. Massen, Wendt Müller, Paul M. Nealen, Andreas Nieder, Katarína Pichová, Laurent Prétôt, John Quinn, Juan C. Reboreda, Amanda R. Ridley, Francisco Ruiz-Raya, Janja Sirovnik, Katie E. Slocombe, Michaela Syrová, Alex H. Taylor, Christopher N. Templeton, Camille A. Troisi, Utuku Urhan, Kees van Oers, Alberto Velando, Frederick Verbruggen, Alizée Vernouillet, Petr Veselý, Auguste M. P. von Bayern, Benjamin A. Whittaker, Vanessa A. D. Wilson.

**Validation:** Rachael Miller, Kai R. Caspar, Jimena Lois-Milevicich.

**Visualization:** Rachael Miller, Kai R. Caspar, Jimena Lois-Milevicich, Carl D. Soulsbury, Stephan A. Reber, Raúl O. Gómez.

**Writing – original draft:** ManyBirds Project, Rachael Miller, Vedrana Šlipogor, Kai R. Caspar, Carl D. Soulsbury, Megan Lambert, Ondřej Fišer, Paula Ibáñez de Aldecoa, Anastasia Krasheninnikova, Connor T. Lambert, Christine R. Lattin, Julia A. Mackenzie, Paul M. Nealen, Carla Pascual-Guàrdia, John Quinn, Alex H. Taylor, Utku Urhan, Vanessa A. D. Wilson.

**Writing – review & editing:** ManyBirds Project, Rachael Miller, Vedrana Šlipogor, Kai R. Caspar, Jimena Lois-Milevicich, Carl D. Soulsbury, Stephan A. Reber, Claudia Mettke-Hofmann, Megan Lambert, Laura M. Biondi, Michael W. Butler, Nicola S. Clayton, Ella B. Cochran, James R. Davies, Dominik Fischer, Ondřej Fišer, Isabelle George, Raúl O. Gómez, Lauren M. Guilette, Katie J. Harrington, Paula Ibáñez de Aldecoa, Oluwaseun S. Iyasere, Melanie G. Kimball, Uta U. König von Borstel, Connor T. Lambert, Christine R. Lattin, Zhongqiu Li, Julia A. Mackenzie, Danna Masri, Jorg J. M. Massen, Paul M. Nealen, Carla Pascual-Guàrdia, Ayushi Patel, Katarína Pichová, John Quinn, Nikola Schlöglová, Eva Serrano-Davies, Katie E. Slocombe, Masayo Soma, Alex H. Taylor, Christopher N. Templeton, Camille A. Troisi, Utku Urhan, Kees van Oers, Frederick Verbruggen, Jorrit W. Verkleij, Alizée Vernouillet, Auguste M. P. von Bayern, Ella R. Williamson, Vanessa A. D. Wilson.

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
