## [Editor Report · Decision Letter 0]

4 Dec 2024

Dear Dr Miller,

Thank you for submitting your manuscript entitled "Evolutionary drivers of neophobia across the avian clade" for consideration as a Research Article by PLOS Biology.

Your manuscript has now been evaluated by the PLOS Biology editorial staff, as well as by an academic editor with relevant expertise, and I'm writing to let you know that we would like to send your submission out for external peer review.

Once your full submission is complete, your paper will undergo a series of checks in preparation for peer review. After your manuscript has passed the checks it will be sent out for review. To provide the metadata for your submission, please Login to Editorial Manager (https://www.editorialmanager.com/pbiology) within two working days, i.e. by Dec 06 2024 11:59PM.

Kind regards,

Roli Roberts

Roland Roberts, PhD

Senior Editor

PLOS Biology

rroberts@plos.org

---

## [Decision Letter · Decision Letter 1]

24 Feb 2025

Dear Dr Miller,

Thank you for your patience while your manuscript "Evolutionary drivers of neophobia across the avian clade" was peer-reviewed at PLOS Biology. Your manuscript has been evaluated by the PLOS Biology editors, an Academic Editor with relevant expertise, and by three independent reviewers.

As you will see in the reviewer reports, which can be found at the end of this email, although the reviewers find the work potentially interesting, they have also raised a substantial number of important concerns. Based on their specific comments and following discussion with the Academic Editor, it is clear that a substantial amount of work would be required to meet the criteria for publication in PLOS Biology. However, given our and the reviewers' interest in your study, we would be open to inviting a comprehensive revision of the study that thoroughly addresses all the reviewers' comments. Given the extent of revision that would be needed, we cannot make a decision about publication until we have seen the revised manuscript and your response to the reviewers' comments. Your revised manuscript would need to be seen by the reviewers again, but please note that we would not engage them unless their main concerns have been addressed.

You'll see that reviewer #1 says “the study overcomes an important short-coming in research addressing questions of this kind, namely, species specificity,” but notes a number of significant issues. The first is that the sheer breadth means that it’s much less focussed. S/he also thinks that the lack of clear hypotheses runs the risk of p-hacking and multiple testing problems, that the paper may be better presented as multiple studies (please do not do this - see my note below), that the fact that 90% of the birds are in captivity deserves significant comment, that the degree of consistency between species is unclear, and there is no surprising take-home. Reviewer #2 is more positive, saying, “The scope and ambition of this work are commendable, and the contribution has significant potential.” S/he suggests removing the first part of the Intro, spelling out the research goals more explicitly (including more alternative hypotheses), including better connection between the results and the hypotheses. S/he also suggests changes to the analyses – include control latencies as a covariate, perform log-transformations, capitalise more on the potential of your Bayesian framework, and better clarify/justify your analytical choices. S/he also attached a marked up PDF. Reviewer #3 raises several concerns, some of which that overlap with the other reviewers, and says that in order to address your stated research questions, you need to better account for phylogeny and a range of life history trait variables. S/he also questions the connection between results and stated hypotheses (see reviewer #2).

Having discussed the reviews with the Academic Editor, we think you should try to make the framing of the paper more coherent, including clear up-front hypotheses. There are also a number of specific requests for additional analyses and clearer defence of current analytical choices. The captive nature of most of the subjects should be clearly flagged as a limitation. IMPORTANT: After discussion with the Academic Editor, we encourage you NOT to split up the manuscript, as one of the reviewers suggests ("A split is unnecessary and would diminish the impact of the results. I think the results are interesting enough to justify allowing the authors to revise and respond to the comments").

We appreciate that these requests represent a great deal of extra work, and we are willing to relax our standard revision time to allow you 6 months to revise your study. Please email us (plosbiology@plos.org) if you have any questions or concerns, or envision needing a (short) extension. At this stage, your manuscript remains formally under active consideration at our journal; please notify us by email if you do not intend to submit a revision so that we may withdraw it.

**IMPORTANT - SUBMITTING YOUR REVISION**

*Resubmission Checklist*

*Published Peer Review*

*PLOS Data Policy*

*Blot and Gel Data Policy*

Sincerely,

Roli Roberts

Roland Roberts, PhD

Senior Editor

PLOS Biology

rroberts@plos.org

REVIEWER' COMMENTS:

Reviewer #1:

This paper is the product of a large-scale study, on 136 bird species by 129 collaborators - the ManyBirdsProject. The coordination of such a study to the point of manuscript submission is extremely impressive and the authors deserve congratulations for pulling it off. The research question being addressed is - what are the evolutionary drivers of neophobia? - and the hypotheses being tested range from phylogenetic to social behaviour, migratory behaviour, ecology and diet. Each species in the study was tested for neophobic tendencies with the same experiment to allow for comparison across species.

Generally speaking, the study overcomes an important short-coming in research addressing questions of this kind, namely, species specificity. The authors note that when there have been attempts to compare neophobic behaviours across species, comparisons have been restricted to a group of species within the same family, for example, crows. While there are comparative methods for comparing data from across species - meta-analysis, for example - the authors here have gone one step further and organise efforts to replicate the same experiment; the theory being that this will provide directly comparable data. So, along with the effort, the aims are laudable, and the study has identified some clear patterns in the data.

The trade-off is that the bigger your study, the less focussed it is, the greater the risk of missing important details, eg species differences; the more you need to think about the risks of multiple comparisons (the need for Bonferroni correction for example).

I've listed some points below that the authors may wish to consider or clarify:

- I wasn't clear to me whether hypotheses were tested a priori, and if so, what those were. This is so important in a study like this because of the risk of p-hacking accusation and failing to account for multiple comparisons.

- There may just be too much in this for one paper: any one of ecology, sociality, phylogeny, diet could be publishable alone - putting them all together is a little over-whelming and precludes in-depth examination of the data.

- I was astonished that 90% of the subjects were captive birds -this seems extremely high and I wanted to understand the implications of this a little more. To what extent is captivity expected to impact on neophobia and are some species likely to be impacted more than others: raptors vs budgies?

- The experimental set-up, picture in Figure 7 made me wonder if the same thing really was tested across 136 species - is it a risk that some species will react to these objects differently from others in a way that obscures underlying neophobia, frugivores may be more comfortable with bright colours for example.

- I needed some help from the authors to assess the significance of the findings. None of them seemed that surprising to me: neophobia has a phylogenetic signature (few traits don't); generalists are less neophobe than specialists.. did we need this study to tell us that? I was missing a sense of "now we know!" which you might expect from a study on this scale.

Reviewer #2:

This manuscript represents an impressive effort to compile high-resolution data on an important behavioral feature—neophobia—across a wide range of species, aiming to identify the socio-ecological drivers of evolutionary variation. The scope and ambition of this work are commendable, and the contribution has significant potential to advance our understanding of the adaptive nature of neophobic responses.

However, there are several aspects where the manuscript could be improved to strengthen its clarity, logical flow, interpretation of results, and overall impact. Below, I outline specific suggestions for improvement, organized into conceptual issues and methodological suggestions:

1. Conceptual Issues

First, the initial focus on behavioral flexibility in the introduction seems unclear and unnecessary, given that this link is not revisited later in the discussion. The central argument of the manuscript is that neophobia is stable enough for meaningful comparisons across species. Therefore, I suggest removing the first section of the introduction discussing flexibility and instead beginning directly with a definition of neophobia and its ecological and evolutionary significance. This would sharpen the manuscript's focus.

The research goals should be reorganized for better logical structure. The first step is to test for individual differences in neophobia, as this measures repeatability and directly assesses plasticity. This approach demonstrates that, despite substantial plasticity, there is a stable component within individuals. Note however that even if no stable component were found, species-level differences in neophobia would not be negated. The second step is to actually assess differences across species. This evaluates whether species differ significantly in neophobia despite plasticity (i.e. neophobia is a species feature), a major assumption of the comparative approach. Including phylogeny as a random effect can further help assess to what extent differences across species arise from shared ancestry (i.e. ravens are neophobic because they are corvids and hence share a common ancestor and similarities in ecological lifestyle). Finally, the third step is to evaluate whether these taxonomic differences are driven by socio-ecological factors. This is the most significant contribution of the manuscript but should logically follow the validation of interspecific differences in neophobia.

The theoretical framework could be expanded to include additional hypotheses. Beyond the Neophobia Threshold Hypothesis (NTH) and Dangerous Niche Hypothesis (DNH), the manuscript could incorporate life history theory, which predicts that species with slow life histories (e.g., long reproductive lifespans) are risk-averse, while those with fast life histories (e.g., high fecundity) are likely risk-takers. This framework could explain substantial variation in neophobia and help interpret related results, such as the relationships between body size, innovativeness, invasiveness, and migration. Relevant work, such as studies on fear responses to humans, could be referenced (e.g., https://doi.org/10.1007/s00265-018-2463-0). Another missing hypothesis is the null hypothesis that neophobic responses are highly plastic, with no consistent differences across species. This would explicitly test whether species differences reflect inherent traits or environmental flexibility. On a side note, as the NTH and DNH are heavily based on Russell Greenberg's work, adding his name before the citation could be a meaningful homage.

The logic of the specific hypotheses tested in the study needs clearer explanation. While Table 1 outlines these hypotheses, it is not immediately clear what predictions they make or how they relate to the broader theoretical frameworks offered by the NTH and DNH. For example, it is unclear why body size or territorial behavior should influence neophobia. Some of these connections only become clear in the discussion, while others remain unresolved. Including these predictions directly in Table 1 would help clarify their rationale and strengthen the manuscript's presentation.

The manuscript needs stronger connections of the specific hypotheses of Table 1 and the NTH and DNH. Without this, it is unclear how the results support these theoretical frameworks. After reading the manuscript, I'm still unsure to what extent the specific hypotheses allow to disentangle between the NTH and DNH. For example, being an island dweller may reduce neophobia, as shown for parrots by Mettke-Hoffman. This is consistent with both the NTH and DNH, as islands harbour fewer predators and provide ecological opportunities to should reduce neophobia. The authors do not test this particular hypothesis, perhaps due to the lack of data, but the issue I illustrate may apply to many of the specific hypotheses presented in Table

2. Methodological Suggestions

Regarding the response variables, the authors currently use two approaches: the difference between treatment and control conditions, and the raw latency data for each condition. However, variation in the latency to touch the food within and across species can be greatly influenced by motivation to feed. For example, species with higher metabolic rates need to forage more frequently, which may be reflected in shorter latencies for both the treatment and control. This makes some results with raw data difficult to interpret. A better approach would be to include the control latencies as a covariate in the model. Additionally, latency data often deviate from normal distributions, even after log-transformation, and care should be taken to select an appropriate distribution to model it (the BRMS package in R offers some distributions that can be useful here).

On the other hand, differences between treatment and control should be log-transformed to make them proportional and less affected by latency magnitude. Furthermore, the order of treatment and control presentation should be included as a fixed effect in the model (see justification below).

The Bayesian framework employed in the manuscript, specifically the use of MCMCglmm, is a strong methodological choice but has not been fully utilized. Analyzing neophobia at the individual level allows for the estimation of intra-class coefficients, measuring variance across individuals, species, and phylogenetic levels simultaneously. This approach could include covariates such as sex and age, avoid reliance on taxonomic levels to assess phylogenetic effects, and provide direct measures of phylogenetic heritability. It would also help distinguish between variance explained by phylogeny versus species-specific idiosyncrasies, further refining the analysis.

The current approach of adding all predictors into a single model also risks obscuring relationships due to collinearity, confounding effects, or varying measurement errors. For example, testing the effects of urban environments and generalist foraging behavior in the same model may yield misleading results if there is a causal relationship between them. If generalist foraging predicts urban success, which in turn influences neophobia, modeling them simultaneously may mask these causal pathways. A better approach would involve constructing directed acyclic graphs (DAGs) to represent the causal relationships among predictors, followed by testing these relationships using phylogenetic path analysis. Alternatively, the authors could use a model selection approach, testing different combinations of predictors and identifying the models that best explain variation in neophobia.

Important details of the analytical methods are missing and should be included. For example, the error structures, priors, and any convergence diagnostics used in the MCMCglmm analyses should be described. Latency data, in particular, often do not fit well with Gaussian models, and this limitation should be addressed explicitly.

Finally, certain aspects of the experimental protocol require better justification. If the control condition is intended to equalize feeding motivation, why is it conducted 48 hours apart? Motivation may fluctuate during this interval, introducing potential confounds. Additionally, exposing individuals to a novel object first may have carry-over effects on subsequent feeding motivation, which could inflate repeatability estimates. Including treatment-control order as a covariate could help address this bias.

In summary, while the manuscript makes a commendable effort to explore neophobia across species, addressing these conceptual and methodological issues will greatly enhance its clarity, rigor, and overall impact. More specific comments are added to the attached pdf version of the manuscript.

Reviewer #3:

Review comments on "Evolutionary drivers of neophobia across the avian clade".

This study explored avian species' neophobia across different species by analyzing the effect of phylogenetic variation and ecological drivers that potentially shape species variation of neophobia. The study used a large data set provided by Big Team Science collaborations, which enabled authors to compare neophobia across the species. The results showed that the neophobia varied among species (phylogenetic variation) and species variation of neophobia was explained by some ecological factors such as diet and habitat characteristics. Using such a large dataset including various species is amazing and the results may provide variable insights into our better understanding of ecological and evolutionary mechanisms. I appreciate authors' huge effort in organizing this big project. At the same time, I also found several serious issues. First, I don't think this study could correctly explore the evolutionary mechanism of neophobia. Second, the study did not take other important life history traits that may play an important role in shaping species variation of neophobia into account. For these reasons, I think this manuscript needs major fundamental revisions. I will provide major comments rather than minor and point-by-point suggestions because the manuscript may change a lot after addressing comments. I hope these comments will be useful.

Major comments;

1. Evolutionary mechanism was not correctly explored

The main aim of this study is to examine the evolutionary mechanism of neophobia. However, as far as I understand from the manuscript, the study could not examine the evolutionary mechanism of neophobia for a number of reasons. First, this study focuses on phylogeny in objective 1. However, the analysis tested the variation of neophobia at the genus level, while the evolutionary pattern was not correctly analyzed. Variation between the genus does not directly mean the evolution of this trait. I am not a specialist of this type of analysis, but there might be some statistical methods to address this type of question. Second, the result from the within-individual repeatability over the shorter period does not fully support the evolutionary mechanism. Repeatability is more likely to be found from a shorter test interval compared to a longer interval (Bell et al. 2009 Animal Behaviour). Since some avian species used in this study may have longer lifespans, so within-few weeks repeatability may reflect their behavioral tendency over the study period. At least, species life span should be considered. Heritability of this trait should support some aspect of microevolution, but may not be a between-species level evolution. In addition, the repeatability was calculated using all data, representing overall neophobia repeatability. This did not consider species variation. So, I am not able to understand the biological meaning of this repeatability.

2. Other key life history traits were not accounted for.

I understand social, diet, and habitat factors are important drivers for neophobia variation. In addition to these, longevity, developmental speed, reproductive strategies (r-K strategies), and other cognitive characteristics such as innovation may also be factors that can be strongly associated with neophobia. However, these variables were not considered in the analysis. Additionally, brain size may also play an important role in neophobia. The risk-taking behavior measured by flight initiation distance was associated with species' brain size, and I assume neophobia which is also a risk-taking behavior may be associated with species brain size throughout influencing species variation of cognitive mechanisms. I would strongly recommend taking into account these key life history variables.

3. Does Neophobia Threshhold Hypothesis explain the evolutionary mechanism of neophobia?

The life history variables were selected based on the two main hypotheses in this study; (1) Neophobia Threshold Hypothesis and (2) Dangerous Nich Hypothesis. While the Dangerous Nich Hypothesis explains how predation pressure acts as an evolutionary driver that shapes neophobia, the Neophobia Threshold Hypothesis explains how neophobia is the driver that shapes other life history traits. Therefore, as far as I understand from this manuscript, the Neophobia Threshold Hypothesis may not explain the evolutionary mechanism of neophobia. Although neophobia can be feedback to shaping dietary and habitat preference, this is not explained in the manuscript.

4. Flexibility vs consistency

Neophobia was explained as one of the flexible behavioural types. However, this study tested repeatability and found significant within-individual consistency at least over the study period. This means that neophobia is not a flexible behaviour. Is neophobia really flexible behaviour? There is also no explanation in the Discussion section about microevolution and how it is related to evolution at the species scale.

Specific comments;

-Statistical analysis; Response variables for statistical analyses were only "latency to touch familiar food", however, "difference between control and test conditions" may be appropriate because it controlled baseline motivation to food conditioning.

In the model for objective 2, what were the fixed effects? I also wonder how the authors tested model validation. Please clarify these in the main text.

More seriously, the repeatability model did not include any fixed effect meaning the repeatability value is unadjusted. Ecological factors should be considered for repeatability.

-L 212; This is not only this part but throughout the manuscript, socio-ecological factors should be reworded. The social factor is one of the ecological factors. On the other hand, "socio-ecological factor" refers to the factors that are related to human society and ecological factors in ecology. I would suggest rewording to life history factors.

L213; What does this sentence mean? Focusing on closely related species can control phylogeny differences between species and can focus on their main factors. This is not an issue.

L223: This sentence can be more simple and merged with the previous paragraph.

L287; I am not able to understand why testing short-term repeatability can estimate the microevolution of neophobia. Considering the much longer life span for most avian species, using repeatability at the shorter time scale (~2 weeks) seems not appropriate.

---

## [Decision Letter · Decision Letter 2]

8 Aug 2025

Dear Dr Miller,

Thank you for your patience while we considered your revised manuscript "Ecological drivers of neophobia across the avian clade" for publication as a Initial Research Submission at PLOS Biology. This revised version of your manuscript has been evaluated by the PLOS Biology editors, the Academic Editor and two of the original reviewers.

Based on the reviews, we are likely to accept this manuscript for publication, provided you satisfactorily address the remaining points raised by the reviewersand the following data and other policy-related requests.

IMPORTANT - please attend to the following:

a) Please make your Title more explicit and declarative. We suggest that you change it to "A large-scale study across the avian clade identifies ecological drivers of neophobia"

b) Please address the remaining requests from reviewer #1.

c) Many thanks for providing the funding information in Table S9. However, these need to be included in the Financial Disclosure statement (I counted maybe 14 funding bodies listed in that table); please do so.

d) Please address my Data Policy requests below; specifically, we need you to supply the numerical values underlying Figs 1 (treefile), 2 (treefile), 3, 4, 5, S1, S2, S3, S4, either as a supplementary data file or as a permanent DOI’d deposition. I note that you already have an associated Figshare deposition, but this seems to mostly contain metadata and raw data. Please could you complete this deposition with the data and code needed to recreate the Figures?

e) Please cite the location of the data clearly in all relevant main and supplementary Figure legends, e.g. “The data underlying this Figure can be found in S1 Data” or “The data underlying this Figure can be found in https://figshare.com/XXXXXXXX

f) Please make any custom code available, either as a supplementary file or as part of your data deposition.

We expect to receive your revised manuscript within two weeks.

*Published Peer Review History*

*Press*

Sincerely,

Roli Roberts

Roland Roberts, PhD

Senior Editor

rroberts@plos.org

PLOS Biology

DATA POLICY:

Regardless of the method selected, please ensure that you provide the individual numerical values that underlie the summary data displayed in the following figure panels as they are essential for readers to assess your analysis and to reproduce it: Figs 1 (treefile), 2 (treefile), 3, 4, 5, S1, S2, S3, S4. NOTE: the numerical data provided should include all replicates AND the way in which the plotted mean and errors were derived (it should not present only the mean/average values).

CODE POLICY

DATA NOT SHOWN?

REVIEWERS' COMMENTS:

Reviewer #1:

The authors have worked hard to address my and the other reviewers' comments and it's great to see a very much improved version of this ms. Table 1 is a great addition, thank you!

The expanded conclusion section (pasted below) will make it easier for readers to identify advances in understanding that result from the study but it is still written so that the only way to make sense of it is to refer to other sections of the paper. I suggest changing statements about "affects" to specific statements about what the effect is….(as highlighted below).

"… our results revealed robust support for predictions of the Neophobia Threshold Hypothesis and Dangerous Niche Hypothesis, in that both dietary breadth and migratory habits were predictors of neophobia.

What do "migratory habits" refer to? Are migratory birds more or less likely to be neophobic?

We also found some evidence for additional effects of territoriality, habitat density and diversity, and domestication impacting neophobic responses.

How? In what direction?

Reviewer #2:

The new version of the manuscript represents a significant improvement over the previous version, both in terms of clarity, theoretical background and scientific rigor. The authors have addressed many of the concerns of the reviewers and editors, and I believe the current version may now be suitable for publication.

While I disagree with certain specific responses provided by the authors, I do not consider these disagreements to be fundamental or detrimental to the overall conclusions of the study. These points mainly reflect differences in interpretation or emphasis rather than flaws in the methodology or reasoning, and they do not undermine the central findings presented in the manuscript.

This study is significant not only for advancing our understanding of neophobia—a key behavioral response to novelty—but also for pioneering an ambitious collaborative research model that investigates cognition through an individual-based, experimentally driven comparative framework.

---

## [Editor Report · Decision Letter 3]

1 Sep 2025

Dear Rachael,

Thank you for the submission of your revised Initial Research Submission "A large-scale study across the avian clade identifies ecological drivers of neophobia" for publication in PLOS Biology. On behalf of my colleagues and the Academic Editor, Gail Patricelli, I'm pleased to say that we can in principle accept your manuscript for publication, provided you address any remaining formatting and reporting issues. These will be detailed in an email you should receive within 2-3 business days from our colleagues in the journal operations team; no action is required from you until then. Please note that we will not be able to formally accept your manuscript and schedule it for publication until you have completed any requested changes.

Sincerely, 

Roli

Senior Editor

PLOS Biology

rroberts@plos.org